# Electrofreezing of liquid water at ambient conditions

Giuseppe Cassone[1] ✉ & Fausto Martelli [ID][2,3] ✉

Water is routinely exposed to external electric fields. Whether, for example, at physiological conditions, in contact with biological systems, or at the interface of polar surfaces in countless technological settings, water responds to fields on the order of a few V Å$^{-1}$ in a manner that is under intense investigation. Dating back to the 19th century, the possibility of solidifying water upon applying electric fields – a process known as electrofreezing – is an alluring promise that has canalized major efforts since, with uncertain outcomes. Here, we perform long (up to 500 ps per field strength) ab initio molecular dynamics simulations of water at ambient conditions under external electric fields. We show that fields of 0.10 – 0.15 V Å$^{-1}$ induce electrofreezing to a ferroelectric amorphous phase which we term f-GW (ferroelectric glassy water). The transition occurs after ~150 ps for a field of 0.15 V Å$^{-1}$ and after ~200 ps for a field of 0.10 V Å$^{-1}$ and is signaled by a structural and dynamic arrest and the suppression of the fluctuations of the hydrogen bond network. Our work reports evidence of electrofreezing of bulk liquid water at ambient conditions and therefore impacts several fields, from fundamental chemical physics to biology and catalysis.

With at least 20 known crystalline forms and counting, the baroque phase diagram of water is the most complex of any pure substance[1] and is continuously under construction. Two amorphous ices, a low-density amorphous (LDA) and a high-density amorphous (HDA) ice[2], encompass a large set of sub-classes[3]; a third, medium density amorphous ice has recently been proposed[4], while a plastic amorphous ice has been suggested to exist at high pressures[5].

Water is also routinely exposed to external electric fields (EFs). The range of strengths 0.1 – 1 V Å$^{-1}$ is particularly relevant, as it represents the range continuously produced by molecular dipoles fluctuations[6] in aqueous solutions[7–9] and to which water is exposed in countless technological/industrial settings[10]. Recent developments have shown that the reaction rates of common organic reactions can be increased by one to six orders of magnitude upon applying external EFs[11–16], hence paving the way to the adoption of EFs as efficient catalyzers. Comparable EFs, generated by charge separation, endow microdroplets with strong and surprising catalytic power[17–20].

Historically, the possibility of manipulating water kinetics via EFs was first proposed by Dufour in 1862[21]. As experimental techniques matured over the years, such an opportunity became more tangible: the role of EFs on the heterogeneous nucleation of ice in cirrus clouds was addressed in the 1960s[22], and several other investigations followed, starting a vivid scientific debate[23–28]. Recently, Ehre et al.[29] have shown that the kinetics of electrofreezing of supercooled water on pyroelectric materials is highly heterogeneous, favoring the crystallization on positively charged surfaces.

Early – and pioneering – computational investigations based on classical molecular dynamics simulations also joined forces. According to some computational studies based on classical molecular dynamics, static fields in the order of ~ 0.5 V Å$^{-1}$ force liquid water to undergo electrofreezing to crystalline ice[30,31], whereas oscillating fields have the effect of boosting the water molecules' dynamics[32]. On the other hand, ab initio molecular dynamics (AIMD), which account for chemical reactions, and experiments have more recently shown that ~ 0.3 V Å$^{-1}$

[1]Institute for Chemical-Physical Processes, National Research Council, Viale F. Stagno d'Alcontres 37, Messina 98158, Italy. [2]IBM Research Europe, Keckwik Lane, Daresbury WA4 4AD, UK. [3]Department of Chemical Engineering, University of Manchester, Oxford Road, Manchester M13 9PL, UK. ✉e-mail: cassone@ipcf.cnr.it; fausto.martelli@ibm.com

represents a threshold above which water molecules undergo dissociation into oxonium $(H_3O)^+$ and hydroxide $(OH)^-$ ions[33–36]. Seemingly, below this threshold, thermal energy and the associated large intrinsic field fluctuations taking place at the molecular scale impede the ordering of the hydrogen bond network (HBN), a necessary step for crystallization to occur. This task can instead be achieved, according to classical simulations, upon tuning the working pressure to ~5 kPa and imposing external EFs of ~0.2 V Å$^{-1}$[37].

The application of intense EFs to liquid water induces a fast response of water molecules which align their dipole parallel to the field direction[38]. On the other hand, the intrinsic cooperativity of HBs acts as a competing force, slowing down the relaxation and retarding the equilibration times. In this work, we model the response of water to external EFs in the range [0.05 – 0.15] V Å$^{-1}$. We perform extensive AIMD simulations and show that equilibration requires ~150 ps, while previous AIMD studies reached 100 ps or less[38,39]. We show that EFs of 0.10 V Å$^{-1}$ induce a transition to a ferroelectric glassy state that we call f-GW (ferroelectric glassy water) after ~200 ps, while EFs of 0.15 V Å$^{-1}$ require ~150 ps. The transition is signaled by the freezing of translational and rotational degrees of freedom, the suppression of the fluctuations of the HBN, and a drop in the potential energy. Fields of the order of 0.05 V Å$^{-1}$, on the other hand, are not strong enough to induce electrofreezing and the system remains in the liquid phase, although characterized by very sluggish dynamics. To follow the time evolution of water response to external EFs, we probe the system at (non-)overlapping time windows and separately average the quantities of interest (we report, in the Supplementary Information, a comparison between quantities of interest computed at disjoint time windows and averaged over entire trajectories). The length of the observation time windows is crucial to capture the correct physics: too long time windows wash out the information on the non-ergodicity, an issue particularly relevant when the system is drifting towards equilibrium. Too small time windows, on the other hand, sample a limited portion of the configurational space inducing artificial suppression of the first moments of the quantities of interest.

Our work provides evidence of electrofreezing of bulk liquid water occurring at ambient conditions.

## Results

Our results show that equilibration is achieved after ~150 ps in the case of the strongest field of 0.15 V Å$^{-1}$, while it requires ~200 ps for the lower fields of 0.10 – 0.05 V Å$^{-1}$. In this Section, we present results obtained at equilibrium, unless otherwise specified.

### Infrared spectra

In Fig. 1 we report the infrared (IR) spectra for bulk water without field (violet line) and for increasingly higher applied fields (0.05 V Å$^{-1}$, blue; 0.10 V Å$^{-1}$, orange; 0.15 V Å$^{-1}$, red). In the absence of applied fields, the position of the OH stretching band – located at 3220 cm$^{-1}$ – and that of low-frequency libration mode – at 560 cm$^{-1}$ – are in good agreement with the experimental data[40]. Upon exposure of the water sample to external EFs, we observe a contraction of the frequency range ascribed to the vibrational Stark effect[41], as also reported in Ref. 42 and – on limited frequency domains – in Ref. 39. This contraction indicates that the field imposes additional selection rules on the molecular vibrations. The largest frequency shift is associated with the OH stretching band; the corresponding red-shift is in the order of ~75 cm$^{-1}$ each 0.05 V Å$^{-1}$, up to an EF of 0.10 V Å$^{-1}$. However, a milder further red-shift in the order of 35 cm$^{-1}$ occurs at a field of 0.15 V Å$^{-1}$. The red-shift of the OH stretching is generally associated with stronger hydrogen bonds (HBs)[43] and the development of more "ice-like" environments[44]. The reduced magnitude of the relative red-shift upon increasing the field from 0.10 V Å$^{-1}$ to 0.15 V Å$^{-1}$, as quantified by the difference in frequencies reported in the inset of Fig. 1, suggests that the effect of the applied field becomes less intense. Moving towards lower frequencies,

the weak libration+bending combination mode band at 2200 cm$^{-1}$ is commonly associated with the strength of the hydrogen bond network (HBN)[45]. The presence of the external EFs induces an enhancement and a concurrent slight blue-shift of this band, further suggesting that the EF causes a strengthening of the HBN. A similar effect has been reported on the IR spectra of water undergoing supercooling[46] where the strengthening of the HBN is, instead, induced by the reduction of thermal energy. A stronger effect on the vibrational spectrum occurs at lower frequencies, the signature of librational modes. The application of an external EF induces a significant blue-shift and, at 0.10 V Å$^{-1}$ and 0.15 V Å$^{-1}$, the development of a clear new band peaked at ~1000 cm$^{-1}$. This band has been ascribed to the breaking of the isotropy of molecular rotations and the preferential alignment with the field direction[42] and is reported in Supplementary Fig. 1 of the Supplementary Information as well as in several other investigations[38,39,42,47,48].

The picture emerging from the inspection of the IR spectra, therefore, indicates that the EF affects the topology of the HBN in several ways: the red-shift of the OH stretching band occurring upon increasing the applied EF is indicative of a strengthening of the HBs, while the blue-shift of the libration + bending combination mode band and that of the librations at lower frequencies suggests some degree of ordering of the HBN. At the same time, the appearance of a new peak in the librational band indicates an alignment of the molecular dipoles along the field direction.

### Structural properties

The strengthening of the HBs (or their stiffening, as shown in Supplementary Fig. 8 as well as in other first-principles[39] and classical MD[48] simulation studies) and the alignment of the molecular dipoles along the field direction mirror an enhancement of spatial correlations that also persists over time. In order to test this hypothesis, we report, in Fig. 2, the $G_{OO}(r,t)$, the Van Hove correlation function computed between oxygen atoms only. In Fig. 2a we report $G_{OO}(r,t)$ in the absence of external fields. We can observe that weak spatial correlation in the region ~2.8 Å and ~4.5 Å, corresponding to the first and second shells of neighbours, rapidly wear off in timescales of ~5 – 10 ps. The application of a field of 0.05 V Å$^{-1}$, reported in panel b, induces an extension of spatial correlations over slightly longer timescales. Radically stronger responses are induced by more intense fields: a field of 0.10 V Å$^{-1}$ (panel c) and a field of 0.15 V Å$^{-1}$ (panel d) clearly strengthen spatial correlations between ~2–3 Å and 4–5 Å and extend them to timescales above ~35 ps.

By projecting the partial Van Hove correlation functions on the reduced domain constituted by the spatial distances only (i.e., by removing the temporal dependence), we obtain the oxygen-oxygen radial distribution functions $g_{OO}(r)$ that we report in Fig. 3a. Without any applied field (violet), the $g_{OO}(r)$ is that of bulk liquid water with a first peak located at ~2.8 Å and a second peak at ~4.5 Å. Adding a small EF of 0.05 V Å$^{-1}$ (blue) we observe an increase in the intensity of the first and the second peaks with a reduction of the population between the first and second peaks. Upon doubling the intensity of the field and reaching 0.10 V Å$^{-1}$ (orange) we observe an enhanced increase in the intensity of both the first and second peaks and further depletion of water molecules populating the interstitial region. An additional increase in the field intensity to 0.15 V Å$^{-1}$ (red) does not show appreciable changes in the $g_{OO}(r)$ with respect to the previous case, suggesting that no further major structural changes occur in the sample. Our results are in agreement with previous works of some of us[34,42] as well as with the investigations from English and co-workers[39] and Luzar and co-workers[48]. Supplementary Fig. 4 of the Supplementary Information reports the $g_{OO}(r)$ computed in consecutive time windows of 50 ps starting from the beginning of our simulations, hence providing a glimpse of the dynamical structural transformations. In agreement with the profiles of the Van Hove functions, it is possible to observe

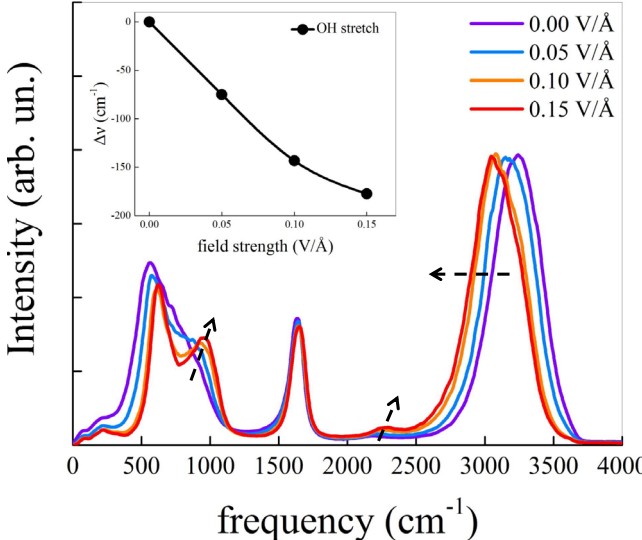

**Fig. 1 | IR spectra under electric field.** Infrared (IR) absorption spectra of liquid water determined at zero field (violet line) and under different field intensities as detailed in the legend. Arrows are guides for the eye qualitatively following the field-induced modifications of the bands. In the inset, we report the vibrational Stark effect of the OH stretching band. A spline function connecting data points has been used.

that the $g_{OO}(r)$ for 0.05 V Å$^{-1}$ converges to the same profile after 50–100 ps (Supplementary Fig. 4a), while convergence is achieved only after 150–200 ps for 0.10 V Å$^{-1}$ (S4-b) and 0.15 V Å$^{-1}$ (S4-c). To verify that the systems reach structural equilibrium at the times mentioned above, we compute the difference between $g_{OO}(r)$s at consecutive time windows. As shown in Supplementary Fig. 5a, the field 0.05 V Å$^{-1}$ induces structural equilibrium after ~100 ps. On the other hand, structural equilibration occurs after ~150 ps for higher fields, as shown in Supplementary Fig. 5b and Supplementary Fig. 5c. We notice, at this point, that the $g_{OO}(r)$ shown in Fig. 3a at fields of 0.10 V Å$^{-1}$ and 0.15 V Å$^{-1}$ at convergence, i.e., after 150 ps of simulation, strikingly resemble the $g_{OO}(r)$ of supercooled water or that of low-density amorphous (LDA) ice[3]. This comparison is, instead, less accurate if one does not take into account the out-of-equilibrium nature that drives the process, and computes the radial distribution functions over the entire trajectories, as reported in Supplementary Fig. 11a as well as in several previous works[34,39,42,48]. To rule out the effect of the simulation box, we have performed longer simulations (up to ~500 ps) for systems with 256 H$_2$O molecules at densities of 0.92 g/cm$^3$ and 0.95 g/cm$^3$. Our results, reported in Supplementary Fig. 9 of the Supplementary Information, show that the development of a glassy-like $g_{OO}(r)$ is independent of the system size and density.

### Dynamical properties

Considering the high computational cost of performing AIMD simulations, we can not produce an equilibrated supercooled sample or an LDA via realistic quenching rates to compare the relative radial distribution functions. Therefore, to understand whether our $g_{OO}(r)$s belong to a glassy sample or a supercooled sample, we look at dynamical properties, namely the diffusivity measured via the mean squared displacement (MSD). Our results are reported in Fig. 3b. It is possible to appreciate how the slope of the MSD drastically drops as soon as we introduce an EF. In the presence of a weak field of 0.05 V Å$^{-1}$ (blue) the sample is still liquid, although the mobility is strongly reduced compared to the case without field (violet). Upon increasing the field to 0.10 V Å$^{-1}$ (orange) and to 0.15 V Å$^{-1}$ (red) the MSD profiles further significantly drop, indicating that water's translational degrees of freedom are confined to molecular vibration, and the rattling within

the cage of the local neighbours. The corresponding diffusion coefficients at the strength of the fields here studied are reported in the inset. Using the diffusivity for the case with no external field as a baseline, we observe a reduction of the diffusion coefficient up to 12.5 times in the case of 0.15 V Å$^{-1}$. Such a substantial drop is several times larger than the field-induced drops reported elsewhere and for similar fields. In particular, Ref. 39 reports a drop of 1.9 times induced by a field of 0.10 V Å$^{-1}$ from AIMD simulations. The reduced drop in diffusivity reported in Ref. 39 is the result of the short timescales there considered, i.e., 100 ps while, as we have mentioned, electrofreezing occurs after ~150 ps. We observe a comparable drop for the case of 0.10 V Å$^{-1}$ if, like in Ref. 39, we limit ourselves to the first 100 ps of simulations. Ref. 48 investigates several classical potentials and indicates that a field of 0.15 V Å$^{-1}$ induces a drop between 1.2 times to 2.5 times depending on the potential. According to Ref. 48, a more substantial drop of ~5.5 times occurs when the EF is increased to 0.20 V Å$^{-1}$, conditions that induce non-negligible water dissociations in our extended AIMD simulations. Therefore, we posit that our enhanced reduction is the effect of performing first-principles simulations on timescales long enough such that they allow for electrofreezing to occur and reach equilibrium. Computing the MSD over wider time windows implies accounting for the contribution of water molecules still out-of-equilibrium, hence artificially increasing the slope of the MSD, as shown in Supplementary Fig. 11b.

To track the slow relaxation on rotational degrees of freedom, we report in Supplementary Fig. 12 the profiles of the rotational autocorrelation functions computed at consecutive time windows for all the cases here investigated and compared against the case without external field. The gradual reduction of rotational degrees of freedom with time, induced by the presence of EFs is clearly visible; eventually, rotational degrees of freedom are fully frozen within the last 50 ps for EFs of 0.10 V Å$^{-1}$ and 0.15 V Å$^{-1}$. Like for the diffusivity, our results on the rotational dynamics in the early stages of the simulations (within 100 ps) are qualitatively in agreement with Ref. 39. Therefore, the freezing of translational and rotational degrees of freedom occurring after ~150 ps is strongly indicative that we are witnessing a dynamic arrest and a transition to f-GW.

### Potential energy

In Fig. 4 we report the profile of the potential energy computed performing single point calculations on 1000 configurations randomly chosen after equilibration is achieved. Panel a reports the profile as a function of the chosen molecular configurations. Without any applied field (violet) the potential energy fluctuates around the dashed violet line. Upon introducing a field of 0.05 V Å$^{-1}$ (blue) we observe a decrease in potential energy for almost all configurations, with an average value (dashed blue line) sitting below the case of water without field. Stronger drops in potential energy occur in the presence of EFs of 0.10 V Å$^{-1}$ (orange) and of 0.15 V Å$^{-1}$ (red). In panel b we report the average potential energy – relative to the zero-field case in kcal/mol – as a function of the field strength. The drop in potential energy is clearly visible and shows how EFs of 0.10 V Å$^{-1}$ and 0.15 V Å$^{-1}$ drag the system into lower potential energy basins[49,50]. It is worth noticing that the reduction in potential energy occurs along with a reduction of ~14% of the entropy, as reported in Ref. 47 for the same system and numerical setups.

### Topology and dynamics of the H-bond network

The amorphization of liquid water involves a sensitive change in the fluctuations and topology of the HBN[51], which can be quantitatively inspected via the ring statistics. Therefore, to confirm that the structural and dynamical changes induced by the EFs indeed prompt a rearrangement in the HBN, we compute $P(n)$, the normalized probability of having a ring of length $n \in [3, 10]$. We here track the changes on the topology of the HBN induced by the external EFs by probing

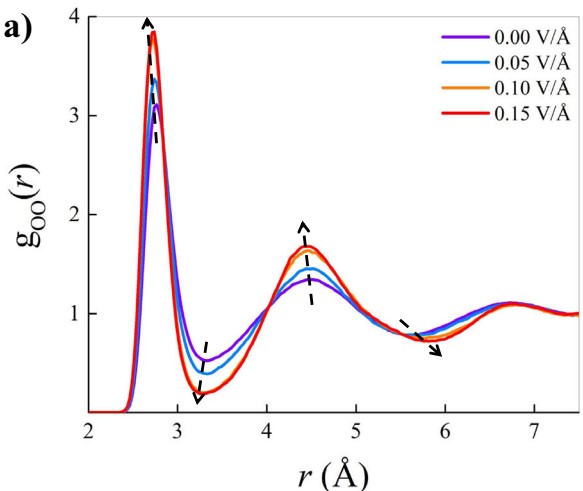

**Fig. 2 | Van Hove functions under electric field.** Partial Van Hove correlation functions between the oxygen atoms (i.e., $G_{OO}(r,t)$) as a function of the time and of the intermolecular distance in the absence of the field (**a**) and in presence of static electric fields with intensities equal to 0.05 (**b**), 0.10 (**c**), and 0.15 V Å⁻¹ (**d**).

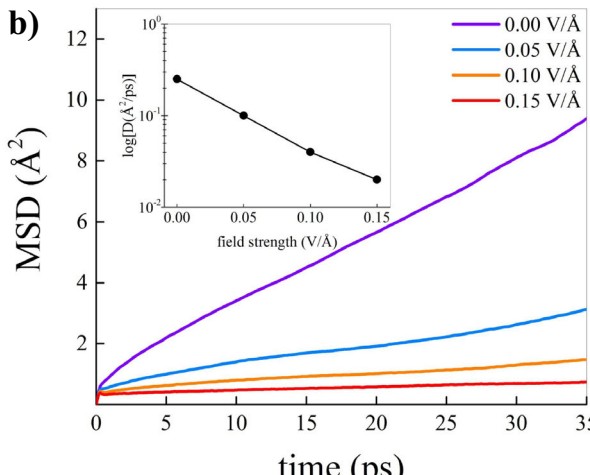

**Fig. 3 | Structural and dynamical properties under electric field. a** Oxygen-oxygen radial distribution functions at different electric field strengths (see legend). Dashed arrows qualitatively depict field-induced modulation of the hydration shells. **b** Mean squared displacement (MSD) of the oxygen atoms at various field intensities (see legend). In the inset, a logarithmic plot of the self-diffusion coefficient of the oxygen atoms as a function of the field strength, where a black solid line connects the data points.

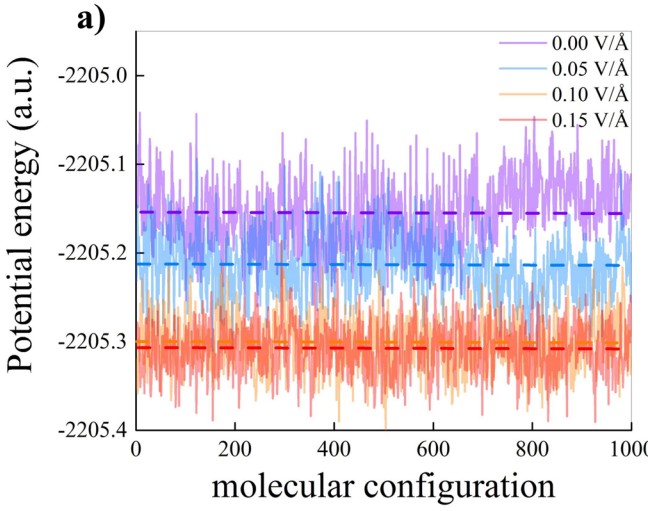
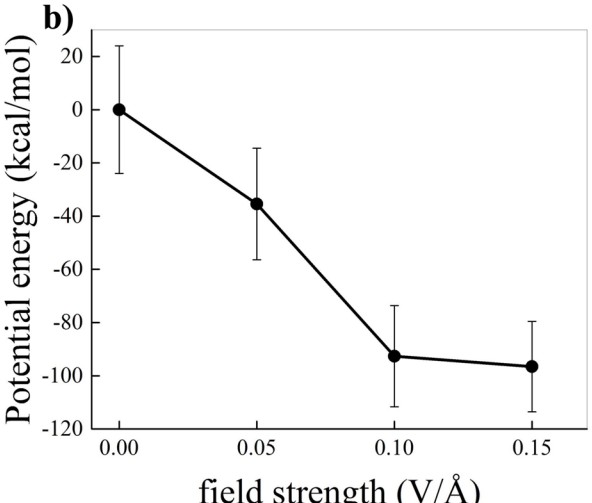

**Fig. 4 | Potential energy under electric field.** Potential energy (in atomic units) computed via single point calculations on 1000 configurations randomly chosen within the time window [201–250] ps of each simulation. **a** Profile of the potential energy in a.u. for water without field (violet), water in the presence of 0.05 V Å⁻¹ (blue), water in the presence of 0.10 V Å⁻¹ (orange), and water in the presence of 0.15 V Å⁻¹ (red). Dashed lines represent the average value. **b** Average potential energy relative to the zero-field case in kcal/mol as a function of the field strength. A black solid line connects the data points and the standard deviation (vertical bars) is shown.

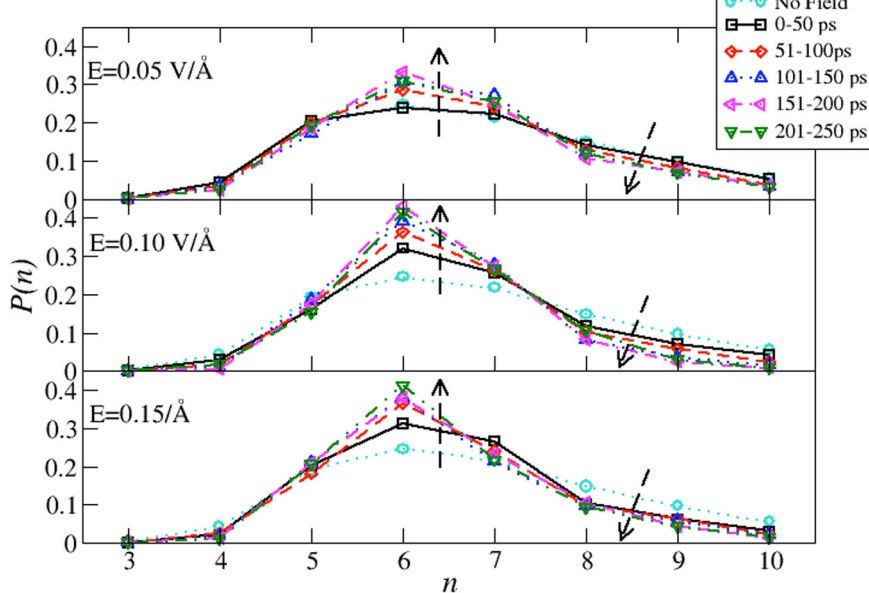

**Fig. 5 | Ring statistics.** Probability distribution $P(n)$ of having a ring of length $n \in [3, 10]$ computed at different time windows during our simulations. The upper panel refers to the applied field $E = 0.05$ VÅ⁻¹, the middle panel to the applied field $E = 0.10$ VÅ⁻¹, the lower panel to the applied field $E = 0.15$ VÅ⁻¹. The cyan circles refer to the zero-field case. The black squares refer to the first 50 ps, red diamonds to the time window 51 – 100 ps, blue upper triangles to the time window 101 – 150 ps, the left magenta triangles to the time window 151–200 ps, and the green lower triangles to the window 201–250 ps. The dashed arrows emphasize the change in $P(n)$ at consecutive time windows.

consecutive time windows of 50 ps until convergence is reached at -150 ps. In hexagonal/cubic ice at 0 K and without defects, the $P(n)$ is centered at $n = 6$, indicating that only hexagons are present. In Fig. 5 we report $P(n)$ for strengths of 0.05 V Å⁻¹, 0.10 V Å⁻¹, and 0.15 V Å⁻¹. Each case is reported against the $P(n)$ determined in the absence of the EF (cyan circles). In the case of 0.05 V Å⁻¹ during the first 50 ps (black squares, panel a), we can observe that the topology of the HBN overlaps almost perfectly with that of liquid water. Upon increasing the simulation time, the topology of the HBN responds to the presence of the field by increasing the number of hexagonal and heptagonal rings while reducing the number of longer rings. Overall, the response of the HBN to the presence of a weak field resembles the transformation of the HBN topology upon cooling[51,52].

Upon doubling the field intensity to 0.10 V Å⁻¹ (panel b), the topology of the HBN drastically changes even within the first 50 ps of simulation. In particular, we observe an increase in hexagonal and heptagonal rings with a corresponding decrease in longer rings. At consecutive simulation time windows, we observe a further sharpening of the $P(n)$ with a considerable increase of hexagonal rings and a depletion of octagonal and longer rings. The topology of the HBN within the last 50 ps of our simulation is remarkably similar to that of LDA (obtained from classical simulations[51]).

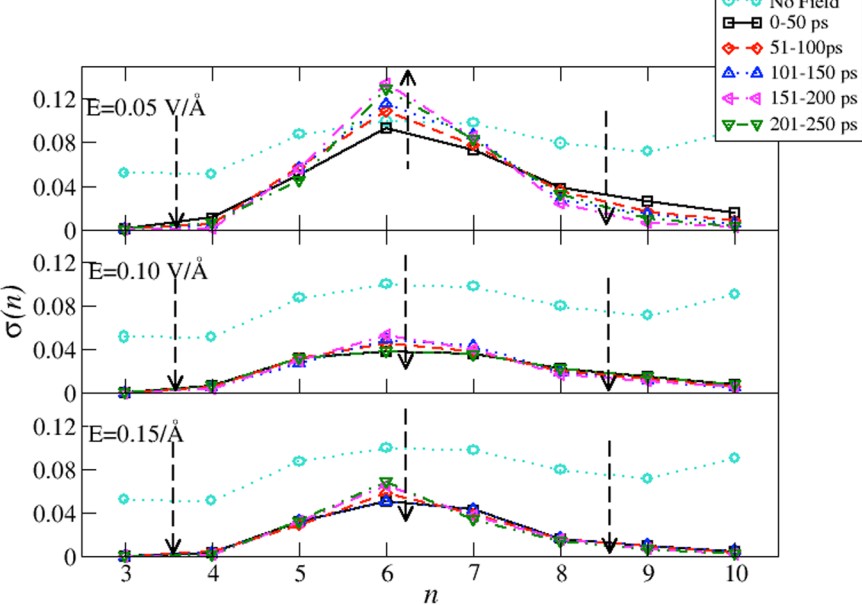

**Fig. 6 | Standard deviation of the ring statistics.** Fluctuations $\sigma(n)$ computed on the ring statistics at different time windows during our simulations. The upper panel refers to the applied field of $E = 0.05$ VÅ$^{-1}$, the middle panel to the applied field of $V = 0.10$ VÅ$^{-1}$, and the lower panel to the applied field of $E = 0.15$ VÅ$^{-1}$. The cyan circles refer to the case of no field. The black squares refer to the first 50 ps, red diamonds to the time window 51–100 ps, blue upper triangles to the time window 101 – 150 ps, the left magenta triangles to the time window 151–200 ps, and the green lower triangles to the window 201–250 ps. The dashed arrows emphasize the $P(n)$ change at consecutive time windows.

A similar behaviour occurs when we apply a field of 0.15 V Å$^{-1}$ (panel c): the HBN reacts to the presence of the field increasing the population of hexagonal and heptagonal rings while decreasing the population of longer rings. Upon increasing the simulation time, the topology of the HBN further increases the population of hexagonal rings while decreasing longer rings, including heptagonal rings.

The gradual rearrangement of the topology of the HBN described above occurs on slower timescales compared to the alignment of water's dipole moment (see Supplementary Fig. 3) and clearly shows that, although single water molecules react very quickly to the presence of EFs, the overall network of bonds reorganizes itself into steady configurations on longer times, as also partially reported in Ref. 53. Such time-dependence can be seen in the gradual build-up of four-coordinated water molecules shown in Supplementary Fig. S10. This gradual build-up in time leads to an increase in four-coordinated molecules up to 15% from the early stages of the simulation. Such an increase in the percentage of four-coordinated environments also induces a gradual enhancement of the local order. We report, in Supplementary Fig. 7, $P(I)$, the probability distribution of the local structure index $I$ estimated on consecutive windows of 50 ps. It is possible to appreciate the development of bimodality in the later stages of our simulations for fields of 0.10 V Å$^{-1}$ (middle panel) and 0.15 V Å$^{-1}$ (lower panel). The lower panel of Supplementary Fig. 7 reports a comparison between $P(I)$ computed in the time window [201 – 250] ps for 0.15 V Å$^{-1}$ and for LDA at $T = 200$ K obtained from classical molecular dynamics simulations. Despite the differences in simulation techniques, the local structure of liquid water under EF strongly resembles that of LDA.

Our investigations on the time evolution of the HBN exposed to external EFs indicate that the systems reach equilibrium configurations after ~150 ps. To confirm this, we compute the difference, at consecutive time windows, of the topology of the HBN. In Supplementary Fig. 13 we report the difference of $n$ − member rings, normalized for the number of water molecules, between consecutive time windows. The upper panel displays the results for 0.05 V Å$^{-1}$ and shows that fluctuations are attenuated at consecutive time windows, but never wear out since the system is still in the liquid phase. The middle and the

lower panels, on the other hand, display the results for 0.10 V Å$^{-1}$ and for 0.15 V Å$^{-1}$, respectively, and clearly show that the HBN converges to steady configurations after ~150 ps.

The information collected so far indicates that our samples gradually readjust to the presence of external EFs. The slow evolution in time involves (i) the gradual development of four-fold configurations interacting via stronger HBs, (ii) the congruent development of more ordered local environments, (iii) the slow reduction of translational and rotational degrees of freedom, (iv) a drop in the potential energy, and (v) the gradual rearrangement of the HBN topology towards configurations richer in hexagonal rings. Eventually, after exposing the samples of liquid water to a field of 0.15 V Å$^{-1}$ for ~ 150 ps, we observe near-complete freezing of translational degrees of freedom, hence suggesting that our sample might be a glass. Although the definition of glassy water is precise (molecular relaxation time exceeding 100 s or the shear viscosity reaching to 1013 poise), our simulations are too short to access these quantities. On the other hand, it has been recently shown that the transition to glass upon quenching liquid water is clearly signaled by the damping in the fluctuations of the HBN topology[51], which we here evaluate and report in Fig. 6 for the three cases in presence of the EF and against the fluctuations computed in liquid water without EF (cyan circles). For all cases, we determine $\sigma(n)$ in time windows of 50 ps. In the case of 0.05 V Å$^{-1}$, we can observe that, with respect to the case in the absence of the field, the fluctuations are strongly damped but for hexagonal and pentagonal rings, which fluctuate in a comparable measure. Upon increasing the simulation time, the fluctuations of the HBN are reduced for all cases but for the hexagonal rings, which become increasingly enhanced with the simulation time. Considering that the sample is liquid (although with strongly reduced diffusion), we posit that the diffusion occurs via changes in the HBN mostly involving hexagonal rings.

At 0.10 V Å$^{-1}$ and 0.15 V Å$^{-1}$, we observe a drastic suppression of the fluctuations of the HBN, to values well below those of the liquid. Such marked reduction of the fluctuations is responsible for the suppression of long-range density fluctuations occurring in correspondence with the transition to glassy water[51,54], a characteristic that

differentiates liquid water from glassy states[55]. Therefore, our findings are strongly indicative of a transition to a glass.

## Discussion

In this work, we have performed long ab initio molecular dynamics (AIMD) simulations of bulk water at ambient conditions in the presence of applied external electric fields (EFs) in the range $0.05 \leq$ EFs $\leq 0.15$ V Å$^{-1}$. The presence of such EFs affects the equilibration times and, therefore, very long simulation times are required to correctly capture the physics of the problem. In the presence of an EF of 0.05 V Å$^{-1}$, the dipoles align along the direction parallel to the EF while the diffusivity becomes sluggish. Overall, the inspected quantities computed within the last ~150 ps of simulation including active rotational and translational diffusivity indicate that the system is genuinely a liquid.

Upon increasing the EF to 0.10 V Å$^{-1}$ and to 0.15 V Å$^{-1}$, we observe a transition to a ferroelectric glass that we call f-GW (ferroelectric glassy water). The amorphization occurs after ~150 ps and is signaled by the freezing of the translational degrees of freedom and a drop in the potential energy, indicating that the sample has reached a metastable basin on the potential energy landscape. The evolution in time of the radial distribution functions and of other structural descriptors report an enhancement of the first and the second shells of neighbours along with a drastic depletion of the entries populating the space between them, as expected in the low-density glassy water. Similarly, the hydrogen bond network (HBN) undergoes a progressive structural reorganization favoring hexagonal motifs and a corresponding suppression of its fluctuations, as expected in the low-density glassy water state[51].

Our work reports evidence of electrofreezing of liquid water at ambient conditions, a task that has been attempted since 1862[21]. The f-GW phase takes a long time to reach and equilibrate and thus requires long (>150 ps) AIMD simulations and is a new tile in the complex phase diagram of water. Therefore, it enriches our understanding of the physics of this complex material. Nonetheless, the conditions explored in this work are ubiquitous in industrial and natural settings, fields that can potentially benefit from this work. For example, water is routinely exposed to natural EFs comparable to the ones explored in this work when at the interface with enzymes, proteins, and biological membranes, defining the biological functionality and stabilizing such complex structures.

We infer that an experimental validation of our finding and the realization of the f-GW phase might be relatively straightforward by exploiting modern experimental settings. Many laboratories are nowadays capable of quantifying the field strengths generated in the proximity of emitter tips[10,13,56] – such as those established by STM and AFM apparatus –, which fall in the same range required to transition to the f-GW. Nonetheless, we posit that lower fields may induce electrofreezing to f-GW on longer timescales, accessible to accurate interaction potentials such as, e.g., MB-Pol[57] or very recently developed Neural Network potentials specifically trained for simulating the response of atomistic systems to external fields[58].

## Methods

### Numerical simulations

We performed ab initio molecular dynamics (AIMD) simulations using the software package CP2K[59], based on the Born-Oppenheimer approach. The external electric fields (EFs) are static, homogeneous and directional (i.e., along the $z$-axis). The implementation of external EFs in Density Functional Theory (DFT) codes can be achieved via the modern theory of polarization and Berry's phases[60–62]. In particular, owing to the seminal work carried out by Umari and Pasquarello[63], nowadays AIMD simulations under the effect of static EFs with periodic boundary conditions are routinely performed. The reader who is interested in the implementation of EFs in atomistic simulations can

refer to the following literature: Refs. [60,61,63–68]. The main simulation here presented consists of a liquid water sample containing 128 $H_2O$ molecules arranged in a cubic cell with side parameter $a = 15.82$ Å, so as to reproduce a density of 0.97 g · cm$^{-3}$. Furthermore, additional simulations were executed on bigger cubic cells composed of 256 water molecules and having edges of 20.05 Å and 20.26 Å. In such a case, lower densities of 0.95 g · cm$^{-3}$ and 0.92 g · cm$^{-3}$ were simulated, respectively. To minimize undesirable surface effects, the structures were replicated in space by employing periodic boundary conditions. We applied static and homogeneous EFs of intensities equal to 0.05 V Å$^{-1}$, 0.10 V Å$^{-1}$, and 0.15 V Å$^{-1}$ from a zero-field condition in parallel simulation runs. The maximum field strength of 0.15 V Å$^{-1}$ was chosen to prevent water splitting known to occur at larger field intensities[33–36,69]. In the zero-field case we performed dynamics of 50 ps whereas, for each other value of the field intensity, we ran dynamics of at least 250 ps. Besides, as for the simulations of the lower-density states only a single field intensity of 0.15 V Å$^{-1}$ was simulated – in addition to the fieldless cases – for time-scales of ~500 ps ($\rho = 0.95$ g · cm$^{-3}$) and ~450 ps ($\rho = 0.92$ g · cm$^{-3}$). This way, we accumulated a global simulation time approaching 2 ns, whilst a time-step of 0.5 fs has been chosen.

Wavefunctions of the atomic species have been expanded in the TZVP basis set with Goedecker-Teter-Hutter pseudopotentials using the GPW method[70]. A plane-wave cutoff of 400 Ry has been imposed. Exchange and correlation (XC) effects were treated with the gradient-corrected Becke-Lee-Yang-Parr (BLYP)[71,72] density functional. Moreover, in order to take into account dispersion interactions, we employed the dispersion-corrected version of BLYP (i.e., BLYP +D3(BJ))[73,74]. The adoption of the BLYP+D3 functional has been dictated by the widespread evidence that such a functional, when dispersion corrections are taken into account, offers one of the best adherence with the experimental results among the standard GGA functionals[75]. It is well-known, indeed, that neglecting dispersion corrections leads to a severely over-structured liquid (see, e.g., Ref. [76] and references therein). Moreover, a nominal temperature slightly higher than the standard one has been simulated in the main simulations to better reproduce the liquid structure (i.e., $T = 350$ K). Furthermore, the additional simulations at lower density regimes were executed at a lower (supercooling) temperature of $T = 250$ K (see the Supplementary Information for the respective results).

Albeit the BLYP+D3 functional represents e reasonably good choice, computationally more expensive hybrid functionals, such as revPBE0, when simulated along with the quantum treatment of the nuclei performs excellently well for water, as demonstrated by Marsalek and Markland[77]. However, since sufficiently large simulation boxes are necessary to track structural transitions, the inclusion of the nuclear quantum effects is beyond the scope of the present work. Moreover, IR absorption line shapes of liquid water (and ice) are overall reproduced remarkably well by standard AIMD simulations, which include by their nature the explicit quantum adiabatic response of the electrons[78]. In addition, the adherence of the IR and of the Raman spectra evaluated by some of us[42] under zero-field conditions with recent experimental results[40,79] justifies a posteriori the classical treatment of the nuclei. As a consequence, the dynamics of ions was simulated classically within a constant number, volume, and temperature (NVT) ensemble, using the Verlet algorithm whereas the canonical sampling has been executed by employing a canonical-sampling-through-velocity-rescaling thermostat[80] set with a time constant equal to 10 fs. IR spectra have been determined using the software TRAVIS[81] (see the Supplementary Information for further information).

### Network topology

In order to probe the topology of the hydrogen bond network (HBN), we employed ring statistics, a theoretical tool that has proven to be

instrumental in investigating the network topology in numerically simulated network-forming materials. The ring statistics is only one of many graph-based techniques to investigate network topologies and, in the case of water, it has helped in understanding the connections between water anomalies and thermodynamic response functions[52,82] as well as the properties of glassy water[54]. We construct rings by starting from a tagged water molecule and recursively traversing the HBN until the starting point is reached or the path exceeds the maximal ring size considered (10 water molecules in our case). The definition of hydrogen bond follows Ref. [83]: two water molecules are bonded if the distance between oxygen atoms is below 3.5Å and the angle $H_1O_1O_2$ is below 35°. We do not distinguish between the donor-acceptor character of the starting water molecule.

## Data availability

Trajectories generated in this study have been deposited and available on the following link: https://zenodo.org/records/10029720?token= eyJhbGciOiJIUzUxMiJ9.eyJpZCI6IjQzOTAwNTk1LTA0NDUtNGIyMS05 ZDQ2LTI2NGJiODM3ZGNlZSIsImRhdGEiOnt9LCJyYW5kb2oiOiJmYjU 5ZmU2YjA2MmU0ZmU5NmQ5OGIzYjNkNWM5NmIwOSJ9.NfJ-Nk5Em Ouk2WKPupaqgahjJVAMvmK_uui-Mr19cf2vKIVTC9DqVqKjjipeKLNAD X06dFhKFRuIvDN7vWkQYA The source data used in this study are available in the Figshare database, Ref. [84].

## Code availability

Custom computer codes used in this work are available from the Authors upon request.

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

## Acknowledgements

G. C. thanks Dr. F. Saija for interesting discussions on electrofreezing. G. C. acknowledges support from ICSC - Centro Nazionale di Ricerca in High Performance Computing, Big Data and Quantum Computing, funded by European Union - NextGenerationEU - PNRR, Missione 4 Componente 2 Investimento 1.4. G. C. is thankful to CINECA for awards under the ISCRA initiative (PTNQE - HP10C69JGX; ELFI-HB - HP10BZLOOF), for the availability of high performance computing resources and support. G. C. acknowledges the European Union (Next-Generation EU), through the MUR-PNRR project SAMOTHRACE (ECS00000022). F. M. acknowledges support from the Hartree National Centre for Digital Innovation, a collaboration between STFC and IBM.

## Author contributions

G. C. and F. M. contributed equally to all stages of the research project.

## Competing interests

The authors declare no competing interests.

## Additional information

**Supplementary information** The online version contains Supplementary Material available at https://doi.org/10.1038/s41467-024-46131-z.

