## [Peer Review File · Nature Communications]

Reviewers' comments:

Reviewer #1 (Remarks to the Author):

Report on manuscript NCOMMS-23-24176

The authors perform long ab initio MD simulations of liquid water under ambient conditions while exposed to electric fields of different strengths. At field strengths between 0.1 and 0.15 V/Å the authors find the structure to “freeze” as measured through both the van Hove function and the mean square displacement. The changes are analyzed in terms of the topology of the hydrogen-bond network as well as the O-O pair-distribution function (PDF) and the trend towards a PDF similar to low-density amorphous ice is noted. The simulations are sound and impressive and I recommend the paper for publication after the authors have addressed the (relatively) minor comments below.

1) Abstract: The authors make a strong final statement in the abstract that their work “paves the way to novel industrial and technological applications”. In the conclusions, however, this is replaced by “fields that can potentially benefit”. I recommend that the authors either be more specific in the conclusions about the applications or less bombastic in the abstract.

2) Page 3, bottom: “...rapidly ware off” should be “...rapidly wear off”

3) Page 7, last paragraph. The statement that the sample “is indeed a glass” is somewhat problematic. The typical definitions of a glass are based on the molecular relaxation time exceeding 100 s (clearly not attainable here) or that the shear viscosity goes to 10¹³ poise. The authors need to better justify the connection between the HBN network and the glass transition. It is clear that the system (at 0.15 V/Å) is frozen on the 35 ps time scale in Fig. 3b, but is it a glass?

4) Page 7, H-bond topology. I would have intuitively assumed that the molecules would tend to align with their dipoles along the field direction (more chain-like) rather than reducing the size of topological rings, enhancing hexagonal rings. This is observed at field-strength 0.05 V/Å (page 10). At the higher field strengths (Fig. S1) the orientation of the dipoles becomes even more enhanced in the field direction. How does that go together with ring formation? Do the authors have any comments on this aspect? And how are the rings oriented relative the field?

5) Page 8. “Dumped” should be “Damped” (two occurrences).

6) Page 10, bottom. It is not clear to me what a “mother simulation” is.

Reviewer #2 (Remarks to the Author):

In this work, the authors employ ab initio molecular dynamics to investigate the effects of intense static external electric fields on bulk liquid water under ambient conditions. This is indeed a highly interesting question both from a fundamental physicochemical point of view, and from an application point of view, with potential impact on a wide variety of fields from electrochemistry, biochemistry, and geochemistry to astrophysics and astrobiology. Novel and interesting findings in this research area are potentially of wide interest. Because of the central importance of water in chemistry, and that of chemistry in the natural sciences, this topic has indeed been the subject of intense research efforts, and occasionally of heated and still ongoing debates.

In this study three static fields were employed, at the strengths of 0.5, 0.1, and 0.15 V/Å. The authors compute and analyze a set of static and dynamic properties: The infrared absorption spectra, the O-O radial distribution functions and their time-dependence (via the Van Hove functions), the translational diffusion coefficients, the ring structures in the hydrogen bond network, and the tetrahedral order parameter. According to the authors, the most important and novel finding is the electrofreezing of water at a threshold field of 0.1 V/Å, or rather in particular,

that this is observed under ambient conditions, as the authors emphasize twice in the abstract. The transition to the nonlinear response regime is already known to happen at ~ 0.1 V/Å [10.1080/14733140500352322].

Although the study, after major revision, would be suitable for publication, it would be much more suited for publication in a journal with more specific focus on physical chemistry. Only a few new insights are provided by the study, or rather some new specific details to what is already known and published. In my opinion this does not justify publication in Nature Communications.

My main criticism is that most of the results are not as novel as they might seem to be as presented in the manuscript. The authors seem to have overlooked some important studies that address exactly the same question, sometimes with almost exactly the same method and the same field strengths, and reach more or less the same conclusions. Sometimes the claim of novelty is explicitly made, and sometimes a finding is reported without due comparison to similar – or identical – findings in already published studies. This also makes it difficult for a reader who is not well acquainted with the directly relevant literature, to put the findings in their proper context, or to distinguish the new results from those which have already been reported.

Another major issue is that the behavior of water is described rather phenomenologically, with hardly any mechanistic microscopic insights at all. Even on a phenomenological level, the analysis is wanting in comparison to what is routinely done in such investigations. No analysis of hydrogen bond (HB) lifetimes or kinetics at all, no anisotropy analysis, all quantities are reported as spherical averages even though the system is highly anisotropic and we are not dealing anymore with isotropic liquid water where spherical averages are perfectly sufficient. No mechanistic explanation of the drop in diffusion in terms of the currently well-understood mechanisms of rotational and translational diffusion in liquid water, their relation to jump-diffusion mechanism and cooperativity, and local structural defects in liquid water.

IR spectra. The authors present vibrational infrared spectra in Fig. 1. Here, the influence of static (and for that matter, oscillating-) fields on the O-H stretching and the librational band has already been reported in 2017 by Futera and English at fields of 0.5, 0.1 V/Å [10.1063/1.4994694]. The strengthening and shortening of hydrogen bonds under static fields, responsible for the red shift of the stretching band, is well-known and has been described in several studies, including those that employ ab initio MD [10.1063/1.4994694; 10.1038/s41598-019-46449-5; 10.1103/PhysRevLett.108.207801]. Out of these previous studies, the only study they cite in this manuscript is the one from Saitta et al., which they cite in a different context (the threshold field for structural breakdown), but not in this section! Here they cite a study by one of the authors from 2019, and another study on supercooled water, but none of the more directly relevant studies that report the same or similar findings. Perhaps the new finding here is the prominent librational peak at 1000 reciprocal cm under 0.15 V/Å, which is interesting, but not very surprising given the known inhomogeneous broadening of the librational peak, as the authors rightly explain.

Molecular alignment. The alignment with the external field is consistent with what has been previously reported with polarizable and nonpolarizable force fields [10.1063/1.5079393] as well as with ab initio MD simulations [10.1038/s41598-019-46449-5]. All models correctly predict the slowing down of diffusion under static fields. Quantitatively, it is also known that non-polarizable force fields underestimate the impact of the field [Jung1999], while polarizable force fields are in better quantitative agreement with ab initio MD [10.1063/1.5079393; 10.1016/s0301-0104].

RDF. Field-induced changes in the O-O RDF and hydrogen bond kinetics have again been previously reported, with essentially the same outcomes as the ones reported here. This can be found for instance in the study of Futera et al. [10.1063/1.4994694], and in the context of force field MD, in the work of Shafiei et al. [10.1063/1.5079393]. The latter work employs static fields with strengths up to 0.2 V/Å, and additionally, offers much deeper insights into the dynamics of the hydrogen bond network, based on the approach of Luzar et al. [10.1038/379055a0]. The latter work is cited in this manuscript, however no attempt has been made to quantify hydrogen bond kinetics, and hence it is not possible to compare the outcomes to those obtained with force fields in [10.1063/1.5079393], which are at least in qualitative agreement with this study. Fig. 2 offers only a rather qualitative and indirect impression of this kinetics.

Freezing of translational motion. In order to establish the glassy nature of the sample, the authors compute the diffusion coefficients. Here, the effects of external electric fields are already well-understood. AC fields enhance diffusion while DC fields suppress it, and this has been reported both using force fields and AIMD [10.1063/1.4994694, 10.1016/j.cplett.2013.07.030]. In both cases, the modification of the diffusion coefficient is due to the same underlying mechanism: rotational coupling. So existing studies have already reported the same findings for water under ambient conditions. The previously obtained values for $\log_{10}D(\text{\AA}^2/\text{ps})$ in these aforementioned studies were -1.1 (ab initio MD, 0.1 V/\AA), -1.05 (BK3 polarizable force field, 0.1 V/\AA), and -1.5 (BK3, 0.2 V/\AA). These values obviously do not correspond to a liquid, and they indicate a drop of D by about one order of magnitude at 0.1 – 0.2 V/\AA, in agreement with the current study. Those studies were published back in 2017 and 2019, and they are not cited here. I am also having difficulty in making a more precise comparison to the current results. The values of D are not reported anywhere, only as points in the plot in the inset inside Figure 3. Obviously there is a missing minus sign on the y-axis. Still, the value for field-free BLYP-D3 should be around 0.2 $\text{\AA}^2/\text{ps}$ which does not seem to agree with the plotted graph. I am not sure if I am not able to correctly read the graph, or there is some disagreement in the numbers.

HB topology. Regarding the analysis of the topology of the hydrogen bonding network, precisely the observed increase in six-membered rings and decrease in seven-membered rings, has been also reported. For example with TIP4P water under static fields, where the onset of structural changes was reported at 0.15 V/\AA [10.1016/s0301-0104(99)00119-6]. Given the already known tendency of non-polarizable force fields to underestimate the impact of static fields on translational diffusion [See for example /10.1063/1.5079393], TIP4P water remains a liquid while exhibiting these changes in the hydrogen bond network topology. It is unfortunate that the authors do not provide such comparisons and insights, which would surely substantially improve the manuscript, and help place the results in a wider context.

A new glassy phase? Taken together, the results are not convincing that we are indeed dealing with a new phase of water. For instance, the trends in the diffusion constant (Fig. 3b) and alignment (Fig. S2) are continuous and do not seem to exhibit any discontinuities or deviations from the trends under weaker field. As already mentioned, changes in the topology of the hydrogen network similar to what is being reported, have also been observed with liquid water, and the fact that these changes are also similar to those in LDA ice does not in itself establish much.

Potential energy. The reported trend in the potential energy (PE) of the system in Fig. 4 is indeed interesting. To be specific, the lowering of the PE in the presence of the field is not surprising, what is interesting is the levelling of this drop starting with the field strength 0.1 V/\AA. The authors argue that this is evidence that a new phase is forming. But given the trend in the polarization of the system shown in Fig. S2b, this seems to be just a manifestation of the onset of non-linear dielectric behavior. Furthermore, explanation of potential energy changes can be very subtle, describing this as merely a dragging of the system in a lower energy basin does not provide much insight beyond what is already visible in Fig. 4. In liquid water, external fields couple both to molecular rotations and translations [See for example the analysis in 10.1126/sciadv.aay7074]. The interaction of the field with molecular rotations is given by the torque acting on the dipoles. Here one needs to use the dynamic dipoles, given by the Born effective charges, rather than the static dipole defined by the first moment of some molecular charge distribution, the latter are presumably the ones shown in Fig. S2 (both quantities are the same for an isolated molecule, but different in condensed phase).

A back of the envelope calculation: With a value of 2.1 D for the dynamic molecular dipole of water under ambient conditions [10.1038/s41598-019-46449-5] a field of 0.05 V/\AA gives a potential well of ~ -0.6 Kcal/mol/molecule, or -77 Kcal/mol or ~ 0.1 Hartree for the entire simulation box. Given the partial alignment of the molecules with field, the average angle of 30 degrees (Fig. S1) explains the ~ 0.05 Hartree drop in the potential energy of the system. Similarly, the further drop in potential energy at 0.1 V/\AA can be explained. The rather weak decrease in potential energy at 0.15 V/\AA is probably a manifestation of nonlinearity in polarization of the sample (Fig. S2b, but taken with caution as this is the static dipole), this needs further scrutiny but would surely be very interesting to investigate. The role of field coupling to translational motion (specifically the O-O stretch motion at 200 cm^{-1}) is also very interesting, given the freezing of the translational motion.

There are several minor issues that the authors need to address to improve the quality of the

manuscript:

- Why did they choose a density of 0.97 g/cm³ for liquid water? This is neither the density of liquid water at room temperature, nor of hexagonal ice, or even LDA ice. The chosen density might have a very strong impact on all the reported values, as this corresponds to a huge difference in pressure, this point needs to be carefully explained.
- Many details in the computational details are missing:
 - How were the IR spectra computed? Merely citing Travis, an analysis tool, is not enough. If the authors used CP2K to compute the maximally localized Wannier centers, this should be reported. The lengths of trajectory segments used, and the frequency of sampling the trajectory, should all be reported.
 - How were the molecular dipoles in Fig. S2b were computed? Are these based on the positions of the maximally localized Wannier centers? I could not find these details anywhere.
- The coupling constant of 10 fs for the CSV thermostat seems to be too tight. The default coupling constant in CP2K is two orders of magnitude larger (i.e. weaker coupling to thermostat). The general practice is to use a weak coupling so that the dynamics is not strongly perturbed. Granted, CSV is more forgiving in this aspect, but it could be that the authors observed a heating of the sample that could only be controlled by strong thermostating, if this is the case, this should be reported.
- The time-dependence of the total system dipole, or of the average angle between molecular dipoles and field, should be reported for each field strength. The dielectric saturation threshold of water is ~ 0.005 V/Å, and one might observe a small drift in the alignment for simulations in the sub-nanosecond regime.
- Surface effects are not “not physical” (page 11), they are just undesirable in this context.
- There are several omissions in the analysis. In general, the authors tend to describe what is happening with words and in a qualitative manner, rather than with numbers. This does not help when comparing to other studies. For instance:
 - What is the hydrogen bond density? The number of HB per water molecule is an extremely interesting quantity in such studies and should be reported. In the field values employed by the authors, it is already well known that the molecules diffusively and cooperatively align with the field without impacting the HB density.
 - The field introduces a strong anisotropy into the system, reporting the isotropic average of the diffusion coefficient, or of the radial distribution function or HB lengths, as done in equilibrium water, does not provide a full picture. Properties should be reported both in field direction and in the orthogonal plane.
 - The authors correctly point out that the rotational motion of the molecules freezes, maybe it would be a good idea to compute the rotational autocorrelation function, say in the last 50ps? Even better: in different trajectory segments. Again, this is rather standard practice in such studies.
- Error bars are never reported for any of the fluctuating microscopic quantities.

In the following, we report a point-by-point response to the Reviewers. We color in blue our response, in *italic* the modification to the manuscript, and in underscore italic the part of the texts that are relevant part of the original manuscript.

Reviewer #1

The authors perform long ab initio MD simulations of liquid water under ambient conditions while exposed to electric fields of different strengths. At field strengths between 0.1 and 0.15 V/Å the authors find the structure to “freeze” as measured through both the van Hove function and the mean square displacement. The changes are analyzed in terms of the topology of the hydrogen-bond network as well as the O-O pair-distribution function (PDF) and the trend towards a PDF similar to low-density amorphous ice is noted. The simulations are sound and impressive, and I recommend the paper for publication after the authors have addressed the (relatively) minor comments below.

We thank the Reviewer for their highly positive evaluation of our work.

1) Abstract: The authors make a strong final statement in the abstract that their work “paves the way to novel industrial and technological applications”. In the conclusions, however, this is replaced by “fields that can potentially benefit”. I recommend that the authors either be more specific in the conclusions about the applications or less bombastic in the abstract.

We have adjusted the Abstract, the last paragraph now reading as:

Our work represents the first evidence of electrofreezing of liquid water at ambient conditions and therefore impacts several fields, from fundamental chemical physics to biology and catalysis.

2) Page 3, bottom: “...rapidly ware off” should be “...rapidly wear off”

We have corrected this typo.

3) Page 7, last paragraph. The statement that the sample “is indeed a glass” is somewhat problematic. The typical definitions of a glass are based on the molecular relaxation time exceeding 100 s (clearly not attainable here) or that the shear viscosity goes to 10¹³ poise. The authors need to better justify the connection between the HBN network and the glass transition. It is clear that the system (at 0.15 V/Å) is frozen on the 35 ps time scale in Fig. 3b, but is it a glass?

We agree with the Reviewer that we do not provide a quantitative proof based on viscosity of relaxation times, that our system is a glass. Nonetheless, considering the freezing occurs gradually over time and that the applied field is static, we are very confident that we have reached a glassy state. In the revised manuscript, we provide a more thorough connection between the fluctuations

of the HBN and the transition to glass. We have added the following text to the Results Section, page 8 and page 9:

The information collected so far indicates that our samples gradually readjust to the presence of external EFs. The slow evolution in time involves (i) the gradual development of four-folded configurations interacting via stronger HBs, (ii) the congruent development of more ordered local environments, (iii) the slow reduction of translational and rotational degrees of freedom, (iv) a drop in the potential energy, and (v) the gradual rearrangement of the HBN topology towards configurations richer in hexagonal rings. Eventually, after exposing the samples of liquid water to a field of 0.15 V/\AA for $\sim 150 \text{ ps}$, we observe a complete freezing of translational degrees of freedom, hence suggesting that our sample might be glass. Although the definition of glassy water is precise (molecular relaxation time exceeding 100 s or the shear viscosity reaching to 10^{13} poise), our simulations are too short to access these quantities. On the other hand, it has been recently shown that the transition to glass upon quenching liquid water is clearly signaled by the damping in the fluctuations of the HBN topology [49], which we here evaluate and report in Fig. 6 for the three cases in presence of the EF and against the fluctuations computed in liquid water without EF (cyan circles).

We have also computed the rotational autocorrelation functions, that we report in the plot below for the three fields and at consecutive time windows. It is possible to observe how water molecules are rotationally frozen in the last time window of 0.10 V/\AA and 0.15 V/\AA . This observation further strengthens our claims of being in the presence of a glass. We are currently working on a more detailed analysis on this and on the (de)coupling with translational degrees of freedom for a forthcoming manuscript.

4) Page 7, H-bond topology. I would have intuitively assumed that the molecules would tend to align with their dipoles along the field direction (more chain-like) rather than reducing the size of topological rings, enhancing hexagonal rings. This is observed at field-strength 0.05 V/\AA (page 10). At the higher field strengths (Fig. S1) the orientation of the dipoles becomes even more enhanced in the field direction. How does that go together with ring formation? Do the authors have any comments on this aspect? And how are the rings oriented relative the field?

Water molecules indeed respond to the presence of the field by aligning their dipole to the direction parallel to the field, and such response is very rapid. On the other hand, the cooperativity of the HBs slows down the relaxation and interferes with the alignment of the dipoles, creating a three-dimensional network of bonds. To prove this point, we have computed the angle θ formed by the field direction and the dipole moment of each water molecule. With respect to, e.g., the topology of the HBN, which changes drastically over time, the angle does not show major changes after $\sim 50\text{ps}$. The plot of the $P(\theta)$ distributions is here reported and is the new Fig. S3 in the SI:

We have also added the following sentence to the Introduction:

The application of EFs to liquid water induces a fast response of water molecules which align their dipole parallel to the field direction. On the other hand, the intrinsic cooperativity of HBs acts as a competing force, slowing down the relaxation and, in turn, driving the sample out of equilibrium.

We have also added the following statement to the main manuscript:

The gradual rearrangement of the topology of the HBN described above occurs on slower timescales compared to the alignment of water's dipole moment (see Fig. S3) and clearly shows that, although single water molecules react very quickly to the presence of EFs, the overall network of bonds reorganizes itself into new steady configurations on longer times, as also partially reported in Ref. [51].

5) Page 8. “Dumped” should be “Damped” (two occurrences).

We have corrected this typo.

6) Page 10, bottom. It is not clear to me what a “mother simulation” is.

We have replaced “mother” with “main”.

Reviewer #2

In this work, the authors employ *ab initio* molecular dynamics to investigate the effects of intense static external electric fields on bulk liquid water under ambient conditions. This is indeed a highly interesting question both from a fundamental physicochemical point of view, and from an application point of view, with potential impact on a wide variety of fields from electrochemistry, biochemistry, and geochemistry to astrophysics and astrobiology. Novel and interesting findings in this research area are potentially of wide interest. Because of the central importance of water in chemistry, and that of chemistry in the natural sciences, this topic has indeed been the subject of intense research efforts, and occasionally of heated and still ongoing debates.

We thank the Reviewer for recognizing the widespread relevance of the topics investigated in our manuscript.

In this study three static fields were employed, at the strengths of 0.5, 0.1, and 0.15 V/Å. The authors compute and analyze a set of static and dynamic properties: The infrared absorption spectra, the O-O radial distribution functions and their time-dependence (via the Van Hove functions), the translational diffusion coefficients, the ring structures in the hydrogen bond network, and the tetrahedral order parameter. According to the authors, the most important and novel finding is the electrofreezing of water at a threshold field of 0.1 V/Å, or rather in particular, that this is observed under ambient conditions, as the authors emphasize twice in the abstract. The transition to the nonlinear response regime is already known to happen at ~ 0.1 V/Å [10.1080/14733140500352322].

The Reviewer cites a manuscript reporting methodologies and techniques completely different from ours. In the cited work [10.1080/14733140500352322], the Author performs classical simulations on alternating electromagnetic fields, whereas we focus on *ab initio* simulations on continuous electric fields. Nonetheless, words and concepts related to “structural/phase transition”, “electrofreezing” or “glass” never appear in the mentioned paper; not a single statement is present on the structural transitions induced by the electric field. However, we have cited the suggested work (now Ref. [32]) in the Introduction remarking, however, that it refers to classical molecular dynamics and oscillating fields:

[...] based on classical molecular dynamics. According to these studies, liquid water undergoes electrofreezing to crystalline ice when exposed to external static EFs in the order of ~ 0.5 V/Å [30, 31], and the effects of oscillating EFs have also been investigated [32].

My main criticism is that most of the results are not as novel as they might seem to be as presented in the manuscript. The authors seem to have overlooked some important studies that address exactly the same question, sometimes with almost exactly the same method and the same field

strengths, and reach more or less the same conclusions. Sometimes the claim of novelty is explicitly made, and sometimes a finding is reported without due comparison to similar – or identical – findings in already published studies. This also makes it difficult for a reader who is not well acquainted with the directly relevant literature, to put the findings in their proper context, or to distinguish the new results from those which have already been reported.

We respectfully disagree with the Reviewer. Our manuscript reports 75 references (81 in the revised manuscript) fully covering all the literature of relevance to this investigation. We are aware that the literature is vast and spans decades of experimental and computational approaches. Nonetheless, our approach introduces innovative methodologies and techniques never reported before. Therefore, most of the literature is only marginally relevant. We slice the long trajectories in disjoint time windows, and we analyse quantities of interest in each time window. By adopting this technique, we can follow the response of water in time, and we are able to capture a transition to f-GW occurring after ≈ 150 ps. The Reviewer cites manuscripts in which, instead, averages are ran over the entire trajectories, which are often too short to capture the transition to f-GW. In the revised manuscript, we have quantified the difference in looking at the last 50ps with respect to running averages over 250ps.

The revised manuscript reports new Figures showing averages over the entire trajectories. While the results fully agree with the data reported in the literature to which the Reviewer is referring, they lead to misleading conclusions and hide the transition to f-GW. The following figure, now Fig. S10 in the SI, clearly show how running long averages leads to wrong conclusions both from structural as well as from dynamical standpoints:

We have also further emphasised the novelty of our new approach in the revised manuscript. We have added the following sentence to the Abstract:

While the response of single water molecules is almost instantaneous, the cooperativity of the hydrogen bonds induces slower reorganizations that can be captured by dividing the trajectories in disjoint time windows and by performing analysis on each of them separately.

In the Introduction:

Therefore, in order to follow the response of water, it is necessary to probe the system at (non-)overlapping time windows rather than averaging over the entire simulation, and this is the paradigm we have decided to adopt (we report, in the SI, a comparison between quantities of interest computed at disjoint time windows and averaged over entire trajectories).

And

This comparison is, instead, less accurate if one does not consider the out-of-equilibrium nature that drives the process and computes the radial distribution functions over the entire trajectories, as reported in Fig. S10-a as well as in several previous works.

And

Computing the MSD over wider time windows implies accounting for the contribution of water molecules still in the liquid phase, hence artificially increasing the slope of the MSD, as shown in Fig. S10-b.

And in the Conclusions:

The new f-GW phase can be unveiled only by accessing and isolating late portions of long AIMD simulations and is a new tile in the complex phase diagram of water.

Another major issue is that the behavior of water is described rather phenomenologically, with hardly any mechanistic microscopic insights at all. Even on a phenomenological level, the analysis is wanting in comparison to what is routinely done in such investigations. No analysis of hydrogen bond (HB) lifetimes or kinetics at all, no anisotropy analysis, all quantities are reported as spherical averages even though the system is highly anisotropic and we are not dealing anymore with isotropic liquid water where spherical averages are perfectly sufficient. No mechanistic explanation of the drop in diffusion in terms of the currently well-understood mechanisms of rotational and translational diffusion in liquid water, their relation to jump-diffusion mechanism and cooperativity, and local structural defects in liquid water.

The Reviewer labels our analysis as “routinely done”, and then asks us to perform other classical analysis widely reported in the literature. The observables we investigate provide quantitative evidence in support of our claims because they (i) are probed at disjoint consecutive time windows and (ii) are novel in such context. For instance, one of us has only recently shown that the fluctuations of the hydrogen bond network (HBN) can provide quantitative evidence of the transition to a glass upon supercooling liquid water (Ref. [47] (now Ref. [49]) in the manuscript). This piece of information is all but “routine” and is critical for rationalizing our results but is overlooked by the Reviewer. As mentioned above, our work introduces a novel way of thinking which is probing observables at different, consecutive time windows. Finally, as we further report below in this document, our work is based on the careful study of the HBN and its response with time. It is widely understood that the concept itself of HBN involves cooperativity and embeds information on structural defects, hence making the latter analysis redundant and suitable for a follow up publication in one of the specialized journals that the Reviewer is thinking about. Such connections have been previously reported by one of us in, e.g.,

ACS Nano, 14, 8616—8623 2020;

Carbon, 198, 132—141 2022;

J. Phys. Chem. C 125, 6367—6377 2021;

J. Mol. Liq. 329, 115530, 2021.

The following Figure reports the analysis of the structural defects asked by the Reviewer. In this Figure we are comparing the structural defects at different time windows for the three fields intensities against the case of bulk water without any field. Upon applying a field, the number of

four-coordinated molecules increases with time, being the response more rapid in the case of higher fields. This information is already embedded within the ring statistics.

Nonetheless, we have added Fig. S9 which reports the evolution in time of four-coordinated water molecules:

This figure clearly shows that the build-up of four-coordinated environments is a time-dependent process, further supporting the importance of our approach in analysing the trajectories. The value of 50% represents the value for liquid water in the absence of external fields, and the shaded area represent the corresponding region of fluctuations. We have added the following sentences to the revised manuscript:

Such time-dependence, key in our investigation, can be seen in the gradual build-up of four-coordinated water molecules shown in Fig. S9. This gradual build-up in time leads to an increase in four-coordinated molecules up to 15% from the early stages of the simulation. Such an increase in the percentage of four-coordinated environments also induces a gradual enhancement of the local order. We report, in Fig. S6, $P(I)$, the probability distribution of the local structure index I estimated on consecutive windows of 50 ps. It is possible to appreciate the development of bimodality in the later stages of our simulations for fields of 0.10 V/Å (middle panel) and 0.15 V/Å (lower panel). The lower panel of Fig. S6 reports a comparison between $P(I)$ computed in the time window [201 – 250] ps for 0.15 V/Å and for LDA at $T = 200$ K obtained from classical molecular dynamics simulations. Despite the differences in simulation techniques, the local structure of liquid water under EF strongly resembles that of LDA.

Furthermore, we have inspected the HB kinetics by means of widely employed continuous and intermittent HB autocorrelation functions defined in eq. (4) of the SI. The new plot that we report here and as Fig. S7 in the SI, further confirms all our previous findings concerning the electrofreezing of the HB network and the field-induced suppression of its dynamics.

All these aspects are now commented in the SI from page 5 to page 7:

Fig. S7 shows the continuous (Fig. S7-a) and intermittent (Fig. S7-b) autocorrelation functions $c(t)$ of the H-bonds for different field intensities. Within the intermittent definition of $c(t)$, a given H-bond is allowed to cleave within timescales ≤ 5 fs to account for bond fluctuations. Thus, within this latter fast timescale we always assign to s_{ij} a value equal to 1 when considering the intermittent autocorrelation function (see eq. (4)). The application of a 0.05 V/Å field induces only a relatively moderate – with respect to the zero-field case – slow down of the dynamics of the H-bond network recorded by means of the continuous

c(t) function (Fig. S7-a). Instead, significantly more drastic effects are recorded upon applying fields of 0.10 and 0.15 V/Å. Interestingly, the changes produced on the H-bond network kinetics by these field regimes qualitatively resemble those induced by a sizable (~ 40 K) decrease of the temperature [13]. This is also visible from the intermittent H-bond autocorrelation function displayed in Fig. S7-b. Although the H-bond characteristic time recorded at zero field (violet curve) is extended by the application of a field strength of 0.05 V/Å (blue curve), a visible decay of the intermolecular correlations within the timescales of our simulations is recorded at the latter regime. Instead, a field of 0.10 V/Å (orange curve) and a field of 0.15 V/Å (red curve) clearly strengthen the H-bond persistence over sizably longer timescales. These results are fully consistent with the picture emerging from the partial Van Hove correlation functions shown in the main text (Fig. 2).

IR spectra. The authors present vibrational infrared spectra in Fig. 1. Here, the influence of static (and for that matter, oscillating-) fields on the O-H stretching and the librational band has already been reported in 2017 by Futera and English at fields of 0.5, 0.1 V/Å [10.1063/1.4994694]. The strengthening and shortening of hydrogen bonds under static fields, responsible for the red shift of the stretching band, is well-known and has been described in several studies, including those that employ *ab initio* MD [10.1063/1.4994694; 10.1038/s41598-019-46449-5; 10.1103/PhysRevLett.108.207801]. Out of these previous studies, the only study they cite in this manuscript is the one from Saitta et al., which they cite in a different context (the threshold field for structural breakdown), but not in this section! Here they cite a study by one of the authors from 2019, and another study on supercooled water, but none of the more directly relevant studies that report the same or similar findings. Perhaps the new finding here is the prominent librational peak at 1000 reciprocal cm under 0.15 V/Å, which is interesting, but not very surprising given the known inhomogeneous broadening of the librational peak, as the authors rightly explain.

As we already reported in the original manuscript, IR spectra under the action of static electric fields have already been published. In the original manuscript this was clearly stated:

Upon exposure of the water sample to external EFs, we observe a contraction of the frequency range ascribed to the vibrational Stark effect [38], as also reported in Ref. [39].

We emphasise that Ref. [39] (now Ref. [40]) is a work of one of us and refers the reader to the paper of Futera and English [10.1063/1.4994694] (see Ref. [5] of Ref. [40] therein) here mentioned by the Reviewer. Therefore, we do not claim anything new here. A more careful inspection of our manuscript can surely show that the IR spectra is a contextual introductory tile. However, we clearly mention in our manuscript that the IR spectra reported are computed over the last 50ps of very long *ab initio* molecular dynamics simulations, hence being it itself novel compared to whatever the Reviewer cites:

In Fig. 1 we report the infrared (IR) spectra for bulk water without field (violet line) and for increasingly higher applied fields (0.05 V/Å, blue; 0.10 V/Å, orange; 0.15 V/Å, red) computed over the time window [201-250] ps of the respective trajectories.

Nonetheless, we have further clarified this point in the Introduction:

[...] as also reported in Ref. [40] and – on limited frequency domains – in Ref. [41].

This investigation strategy is essential to correctly identify the transition towards the f-GW structure, an aspect completely neglected by the Reviewer.

Finally, the Reviewer claims that the results reported in [10.1038/s41598-019-46449-5] are relevant to our work. The authors of the cited work use a pulsed squared electric field (of duration equal to 500 fs) whilst we apply a constant electric field. Therefore, such comparison is not justified. Our paper shows exactly the opposite, in that pulsed electric fields cannot lead to structural transitions, unless they last for ~ 3 orders of magnitude longer than those used in [10.1038/s41598-019-46449-5]. This difference is a clear indication that the two methodologies can't be compared. We stress here the fact that the cited work [10.1038/s41598-019-46449-5] never mentions – in any single statement – the possibility of freezing liquid water, the essence of our work.

Molecular alignment. The alignment with the external field is consistent with what has been previously reported with polarizable and nonpolarizable force fields [10.1063/1.5079393] as well as with ab initio MD simulations [10.1038/s41598-019-46449-5]. All models correctly predict the slowing down of diffusion under static fields. Quantitatively, it is also known that non-polarizable force fields underestimate the impact of the field [Jung1999], while polarizable force fields are in better quantitative agreement with an initio MD [10.1063/1.5079393;10.1016/s0301-0104].

Here too, we are not claiming any novelty in the alignment of water molecules with the direction of the field. We would like to mention that one of us has published several papers reporting this observation (see, e.g., the most recent one being *JCP* 158, 184502 2023). The Reviewer is missing the main message and the broader picture not previously reported, which is new: the alignment of single molecules is extremely fast, but the HBN requires time, in a fully dynamical process, to adjust. Based on this idea, we probe different time windows and uncover the electrofreezing to f-GW. This concept is novel and never adopted before. We emphasise it several times in the original version of the manuscript. In the Introduction:

we perform long ($\sim [250-500]$ ps) AIMD simulations and show that EFs in the order of $0.10 \leq EFs \leq 0.15$ V/Å induce a structural transition to a new ferroelectric glassy state that we will call f-GW (ferroelectric glassy water). This transition occurs after ~ 150 ps

In the Results section:

we report the infrared (IR) spectra for bulk water without field (violet line) and for increasingly higher applied fields (0.05 V/Å, blue; 0.10 V/Å, orange; 0.15 V/Å, red) computed over the last 50 ps

and

In Fig. 3 a) we report the $g_{OO}(r)$ computed in the last 50 ps of our simulations

and

Fig. S3 of the SI reports the $g_{OO}(r)$ computed in consecutive time windows of 50 ps starting from the beginning of our simulations, hence providing a glimpse of the dynamical structural transformations

and

We stress here that, like for the $g_{OO}(r)$, the MSD are computed on the last 50 ps of our 250 ps runs

Fig. 5 and Fig. 6 report the analysis of the HBN on different time windows. We also write the topology of the HBN drastically changes even within the first 50 ps of simulation

In the Conclusions we remark

We have inspected the out-of-equilibrium process at disjoint time windows and recorded the results on each window

and

The amorphization occurs after $\sim 150 - 200$ ps

Given the fact that the Reviewer has missed the importance of adopting a time-dependent picture, we have further stressed it in the revised manuscript. We have reported the following figure, now Fig. S3, showing the alignment of the water dipoles with the field direction at different time windows:

We have also emphasised that the alignment of the dipoles occurs on shorter timescales compared to the response of the overall HBN, as shown in Figs. 5 and 6:

The gradual rearrangement of the topology of the HBN described above occurs on slower timescales compared to the alignment of water's dipole moment (see Fig. S3) and clearly shows that, although single water molecules react instantaneously to the presence of EFs, the overall network of bonds reorganizes itself into new steady configurations on longer times, as also partially reported in Ref. [51].

RDF. Field-induced changes in the O-O RDF and hydrogen bond kinetics have again been previously reported, with essentially the same outcomes as the ones reported here. This can be found for instance in the study of Futera et al. [10.1063/1.4994694], and in the context of force field MD, in the work of Shafiei et al. [10.1063/1.5079393]. The latter work employs static fields with strengths up to 0.2 V/Å, and additionally, offers much deeper insights into the dynamics of the hydrogen bond network, based on the approach of Luzar et al. [10.1038/379055a0]. The latter work is cited in this manuscript, however no attempt has been made to quantify hydrogen bond kinetics, and hence it is not possible to compare the outcomes to those obtained with force fields in [10.1063/1.5079393], which are at least in qualitative agreement with this study. Fig. 2 offers only a rather qualitative and indirect impression of this kinetics.

Once again, the Reviewer overlooks the point that the RDFs are computed on the last 50ps of trajectories while the works he/she refers to incorrectly average over longer times. The time evolution of the RDFs was already reported in the original SI file.

Fig. S3 of the SI reports the $g_{OO}(r)$ computed in consecutive time windows of 50 ps starting from the beginning of our simulations, hence providing a glimpse of the dynamical structural transformations

The papers cited by the Reviewers refer either to too short *ab initio* simulations or to averaging procedures executed over much longer times. Therefore, the comparison can't (and shouldn't) be made.

Nonetheless, a careful comparison of our RDFs with the ones reported in the papers cited by the Reviewer shows some clear distinction. Such discrepancies arise from the fact that the RDFs reported in the papers cited by the Reviewer have been obtained upon averaging over the entire trajectories, which are also too short to capture the transition to f-GW. We have made this point clear in the revised manuscript, Results section:

This comparison is, instead, less accurate if one does not take into account the out-of-equilibrium nature that drives the process and computes the radial distribution functions over the entire trajectories, as reported in Fig. S10-a as well as in several previous works.

Finally, in the revised manuscript we compute RDFs upon averaging over the entire trajectory and we emphasise how this procedure, adopted in the literature, misses the formation of the f-GW. See the first figure in the response to Reviewer #2 and the corresponding discussion.

Freezing of translational motion. In order to establish the glassy nature of the sample, the authors compute the diffusion coefficients. Here, the effects of external electric fields are already well-understood. AC fields enhance diffusion while DC fields suppress it, and this has been reported both using force fields and AIMD [10.1063/1.4994694, 10.1016/j.cplett.2013.07.030]. In both cases, the modification of the diffusion coefficient is due to the same underlying mechanism: rotational coupling. So existing studies have already reported the same findings for water under ambient conditions. The previously obtained values for $\log_{10}D(\text{\AA}^2/\text{ps})$ in these aforementioned studies were -1.1 (ab initio MD, 0.1 V/\AA), -1.05 (BK3 polarizable force field, 0.1 V/\AA), and -1.5 (BK3, 0.2 V/\AA). These values obviously do not correspond to a liquid, and they indicate a drop of D by about one order of magnitude at 0.1 – 0.2 V/\AA, in agreement with the current study. Those studies were published back in 2017 and 2019, and they are not cited here. I am also having difficulty in making a more precise comparison to the current results. The values of D are not reported anywhere, only as points in the plot in the inset inside Figure 3. Obviously there is a missing minus sign on the y-axis. Still, the value for field-free BLYP-D3 should be around 0.2 $\text{\AA}^2/\text{ps}$ which does not seem to agree with the plotted graph. I am not sure if I am not able to correctly read the graph, or there is some disagreement in the numbers.

As we reported above, the Reviewer compares our results, related to the last 50 ps of long simulations, with longer time averages of often shorter simulations. Therefore, our results are indeed novel. While they may have been present in previous works or not, their presence has been overlooked in the past. See the first figure to the response to Reviewer #2, and the corresponding comment. Regarding the validity of our data in the zero-field condition, we highlight here that the diffusion coefficient of the oxygen atoms we determine is equal to 0.25 $\text{\AA}^2/\text{ps}$, a value in good agreement with well-known estimates from the dispersion-corrected BLYP functional [Lin et al., JCTC 8, 3902 (2012)] and which accounts for the slightly larger temperature at which we perform our dynamics.

HB topology. Regarding the analysis of the topology of the hydrogen bonding network, precisely the observed increase in six-membered rings and decrease in seven-membered rings, has been also reported. For example with TIP4P water under static fields, where the onset of structural changes was reported at 0.15 V/\AA [10.1016/s0301-0104(99)00119-6]. Given the already known tendency of non-polarizable force fields to underestimate the impact of static fields on translational diffusion [See for example /10.1063/1.5079393], TIP4P water remains a liquid while exhibiting these changes in the hydrogen bond network topology. It is unfortunate that the authors do not provide such comparisons and insights, which would surely substantially improve the manuscript, and help place the results in a wider context.

The increase in hexagonal rings is obvious and, as a matter of fact, we report in the manuscript that it can be seen e.g., upon cooling liquid water. One of us has reported this in Refs. [47,48] (now

Refs. [49,50]) of the manuscript. The novelty in our approach is not the ring statistics per se (which has not been performed in [10.1016/s0301-0104(99)00119-6] as accurately as ours, as we inspect rings of lengths from 3 to 10, while [10.1016/s0301-0104(99)00119-6] studies only 5, 6 and 7 folded rings). The novelty is the damping in the fluctuations. The fluctuations in the topology of the HBN provide a quantitative measure of the transition to a glass, as reported in Ref. [47] (and is absent in [10.1016/s0301-0104(99)00119-6]). In the original manuscript we reported:

Given the thermodynamic conditions of our simulations, for the sample to be in a glassy state the topology of the HBN needs to show damped fluctuations [47]. In Fig. 6 we report $\sigma(n)$, the fluctuations of the network topology for the three cases in presence of the EF against the $\sigma(n)$ computed in liquid water without EF (cyan circles). For all cases, we determine $\sigma(n)$ in time windows of 50 ps.

We have further clarified and emphasised this point in the revised manuscript, which now reads as:

The gradual rearrangement of the topology of the HBN described above occurs on slower timescales compared to the alignment of water's dipole moment (see Fig. S3) and clearly shows that, although single water molecules react very quickly to the presence of EFs, the overall network of bonds reorganizes itself into new steady configurations on longer times, as also partially reported in Ref. [51]. Such time-dependence, key in our investigation, can be seen in the gradual build-up of four-coordinated water molecules shown in Fig. S9. This gradual build-up in time leads to an increase in four-coordinated molecules up to 15% from the early stages of the simulation. Such an increase in the percentage of four-coordinated environments also induces a gradual enhancement of the local order. We report, in Fig. S6, $P(I)$, the probability distribution of the local structure index I estimated on consecutive windows of 50 ps. It is possible to appreciate the development of bimodality in the later stages of our simulations for fields of 0.10 V/\AA (middle panel) and 0.15 V/\AA (lower panel). The lower panel of Fig. S6 reports a comparison between $P(I)$ computed in the time window [201 – 250] ps for 0.15 V/\AA and for LDA at $T = 200 \text{ K}$ obtained from classical molecular dynamics simulations. Despite the differences in simulation techniques, the local structure of liquid water under EF strongly resembles that of LDA. The information collected so far indicates that our samples gradually readjust to the presence of external EFs. The slow evolution in time involves (i) the gradual development of four-folded configurations interacting via stronger HBs, (ii) the congruent development of more ordered local environments, (iii) the slow reduction of translational degrees of freedom, (iv) a drop in the potential energy, and (v) the gradual rearrangement of the HBN topology towards configurations richer in hexagonal rings. Eventually, after exposing the samples of liquid water to a field of 0.15 V/\AA for $\sim 150 \text{ ps}$, we observe a complete freezing of translational degrees of freedom, hence suggesting that our sample might be glass. Although the definition of glassy water is precise (molecular relaxation time exceeding 100 s or the shear viscosity reaching to 10^{13} poise), our simulations are too short to access these quantities. On the other hand, it has been recently shown that the transition to glass upon quenching liquid water is clearly signalled by the damping in the fluctuations of the

HBN topology [49], which we here evaluate and report in Fig. 6 for the three cases in presence of the EF and against the fluctuations computed in liquid water without EF (cyan circles).

A new glassy phase? Taken together, the results are not convincing that we are indeed dealing with a new phase of water. For instance, the trends in the diffusion constant (Fig. 3b) and alignment (Fig. S2) are continuous and do not seem to exhibit any discontinuities or deviations from the trends under weaker field. As already mentioned, changes in the topology of the hydrogen network similar to what is being reported, have also been observed with liquid water, and the fact that these changes are also similar to those in LDA ice does not in itself establish much.

The transition to a glass is of the second order and, therefore, continuous, not discontinuous as suggested by the Reviewer. Furthermore, as previously discussed, the damping in the fluctuations of the HBN upon cooling and reported in Ref. [47] (now Ref. [49]) clearly quantifies the transition to a glass in this case. The transition to a glass follows from basic Physics arguments. A structurally liquid-like system with frozen translational degrees of freedom is a glass. We further proof this upon noticing that the dynamics of the HBN is also frozen and the system belongs to a deeper minimum in the potential energy landscape. This latter concept was previously introduced by Stillinger and later developed by Sciortino. Although we understand that we are not following the canonical definition of glass (relaxation time of 100 s, viscosity of 10¹³ poise), the collection of our observations, enriched with the evidence that the fluctuations of the HBN are dampen as in the glass state (Ref. [49] in the revised manuscript) clearly point towards the evidence of a glass state. The revised SI now reports further evidences in support of our hypothesis, namely the dynamics of the HBN (Fig. S7), the suppression of translational degrees of freedom (Fig. S10-b), and a comparison of local structure indexes (Fig. S6, lower panel and reported below). Besides, as for the field-induced suppression of the rotational degrees of freedom, we prove it in this Letter (see the first figure to the response to Reviewer #1). The figure below shows how the strong similarities between local environments in the f-GW and in the LDA obtained from classical MD and quantified via the local structure index. We emphasise here that, given the different methodologies, this comparison is only qualitative.

See also response to the previous comment for further the modifications to the main manuscript and the SI.

Potential energy. The reported trend in the potential energy (PE) of the system in Fig. 4 is indeed interesting. To be specific, the lowering of the PE in the presence of the field is not surprising, what is interesting is the levelling of this drop starting with the field strength 0.1 V/\AA . The authors argue that this is evidence that a new phase is forming. But given the trend in the polarization of the system shown in Fig. S2b, this seems to be just a manifestation of the onset of non-linear dielectric behavior. Furthermore, explanation of potential energy changes can be very subtle, describing this as merely a dragging of the system in a lower energy basin does not provide much insight beyond what is already visible in Fig. 4.

We do not argue that this is evidence of the formation of the new phase. We just emphasise that this interesting levelling of the Potential Energy Surface starting at 0.1 V/\AA is concomitant with many other evidences already present in the original manuscript (i.e., IR spectra, Van Hove correlation functions, radial distribution functions examined at selected time windows, mean square displacement, HBN fluctuations examined at selected time windows, and many others) and that we have now further supplemented by a series of novel data (Figs. S3, S6 (bottom panel), S7, S9, and S10 in the SI). From the combination of all this quantitative evidence we argue the formation of the novel phase.

In liquid water, external fields couple both to molecular rotations and translations [See for example the analysis in 10.1126/sciadv.aay7074]. The interaction of the field with molecular rotations is given by the torque acting on the dipoles. Here one needs to use the dynamic dipoles, given by the Born effective charges, rather than the static dipole defined by the first moment of some molecular charge distribution, the latter are presumably the ones shown in Fig. S2 (both quantities are the same for an isolated molecule, but different in condensed phase).

We do not use static dipoles. Instead, dipole moments distributions (Fig. S2) are determined directly from the electronic structure and, in particular, from the centres of the Maximally Localised Wannier Functions. Thus, dynamical variations of the dipoles are fully considered from the underlying electron density. This is now explicitly stated in the caption of Fig. S2.

A back of the envelope calculation: With a value of 2.1 D for the dynamic molecular dipole of water under ambient conditions [10.1038/s41598-019-46449-5] a field of 0.05 V/\AA gives a potential well of $\sim -0.6 \text{ Kcal/mol/molecule}$, or -77 Kcal/mol or $\sim 0.1 \text{ Hartree}$ for the entire simulation box. Given the partial alignment of the molecules with field, the average angle of 30 degrees (Fig. S1) explains the $\sim 0.05 \text{ Hartree}$ drop in the potential energy of the system. Similarly, the further drop in potential energy at 0.1 V/\AA can be explained. The rather weak decrease in potential energy at 0.15 V/\AA is probably a manifestation of nonlinearity in polarization of the

sample (Fig. S2b, but taken with caution as this is the static dipole), this needs further scrutiny but would surely be very interesting to investigate. The role of field coupling to translational motion (specifically the O-O stretch motion at 200 cm⁻¹) is also very interesting, given the freezing of the translational motion.

We performed a similar back-of-the-envelope calculation prior to submission and concluded that this does not add relevant information to our findings. As mentioned above, our result must be holistically considered in conjunction with the multiple analyses we performed (i.e., IR spectra, Van Hove correlation functions, radial distribution functions examined at selected time windows, mean square displacement, HBN fluctuations examined at selected time windows, and many others).

There are several minor issues that the authors need to address to improve the quality of the manuscript:

- Why did they choose a density of 0.97 g/cm³ for liquid water? This is neither the density of liquid water at room temperature, nor of hexagonal ice, or even LDA ice. The chosen density might have a very strong impact on all the reported values, as this corresponds to a huge difference in pressure, this point needs to be carefully explained.

We chose 3 different densities, different simulation boxes of diverse sizes (128 and 256 molecules), and different simulation timescales (250 and 500 ps) to make sure that our observations are genuine. Our findings are independent from the simulated densities, box sizes, and simulation lengths provided that dynamical timescales of ≈ 200 ps are afforded. All the information gathered from this computational tour-de-force are reported in the manuscript and in the SI.

Many details in the computational details are missing:

- How were the IR spectra computed? Merely citing Travis, an analysis tool, is not enough. If the authors used CP2K to compute the maximally localized Wannier centers, this should be reported. The lengths of trajectory segments used, and the frequency of sampling the trajectory, should all be reported.
- How were the molecular dipoles in Fig. S2b were computed? Are these based on the positions of the maximally localized Wannier centers? I could not find these details anywhere.

IR spectra and dipole moments are computed as we report in detail in Ref. [39] (now Ref. [40]). See comments above. We have also added the following description to the SI:

Infrared spectra shown in Fig. 1 of the main text have been determined by means of the software TRAVIS [1, 2] from the centres of the Maximally Localised Wannier Functions (MLWFs) [3, 4] calculated on the fly during the ab initio molecular dynamics (AIMD) simulations. Molecular dipoles from MLWFs centers can be determined as:

$$\mu = -2e \sum_i r_i + e \sum_j Z_j R_j \quad (1)$$

where e is the electron charge, r_i is the position vector of the MLWF center i , Z_j is the atomic number of the nuclei j whilst R_j is the position vector of this latter. This way, the IR spectra at the investigated field intensities were computed as the Fourier transform of the molecular dipole autocorrelation function along the last 50 ps of the respective simulation trajectories.

- The coupling constant of 10 fs for the CSVN thermostat seems to be too tight. The default coupling constant in CP2K is two orders of magnitude larger (i.e. weaker coupling to thermostat). The general practice is to use a weak coupling so that the dynamics is not strongly perturbed. Granted, CSVN is more forgiving in this aspect, but it could be that the authors observed a heating of the sample that could only be controlled by strong thermostating, if this is the case, this should be reported.

The default coupling constant in CP2K is two orders of magnitude larger because of a series of reasons, not necessarily proving that this should be the only one to be used. In fact, CP2K is a software package also intended for force-fields molecular dynamics simulations, which typically require larger coupling constants for thermostating. As the Referee her/himself correctly argues, CSVN is quite forgiving on this aspect, and we are fully confident on the fact that our results do not depend on the specific choice of this parameter.

- The time-dependence of the total system dipole, or of the average angle between molecular dipoles and field, should be reported for each field strength. The dielectric saturation threshold of water is ~ 0.005 V/Å, and one might observe a small drift in the alignment for simulations in the sub-nanosecond regime.

The total system dipole for different field strengths can be straightforwardly extracted from the plots we report in Fig. S2 of the SI and, more precisely, from the average molecular dipole moment presented in Fig. S2-b. This can be achieved by simple multiplication of the plotted average dipole value times 128, the number of molecules. As for the time-dependence of the total dipole moment, we have now calculated the time evolution of the distributions of the molecular dipoles with respect to the field axis for all the field strengths here investigated, as reported in Fig. S3 of the SI. Such an additional plot further confirmed all our findings. Finally, the requested average angle between molecular dipoles and the field axis is clearly visible from the (more statistically complete and relevant) distributions reported in Fig. S1 of the SI.

- Surface effects are not “not physical” (page 11), they are just undesirable in this context.

We are grateful for this specification. The sentence has been reworded.

There are several omissions in the analysis. In general, the authors tend to describe what is happening with words and in a qualitative manner, rather than with numbers. This does not help when comparing to other studies. For instance:

- What is the hydrogen bond density? The number of HB per water molecule is an extremely interesting quantity in such studies and should be reported. In the field values employed by the authors, it is already well known that the molecules diffusively and cooperatively align with the field without impacting the HB density.

As discussed above, the number of HBs is embedded within the ring statistics. Nonetheless, as we have reported above, we have computed and reported in the manuscript the behaviour in time of the number of four-coordinated water molecules (Fig. S9 in the SI).

- The field introduces a strong anisotropy into the system, reporting the isotropic average of the diffusion coefficient, or of the radial distribution function or HB lengths, as done in equilibrium water, does not provide a full picture. Properties should be reported both in field direction and in the orthogonal plane.

We are reporting, for the first time, the evolution in time of several quantities, instead of averaging over entire trajectories. Therefore, our results do provide a full picture. In principle, infinite analyses could be done but we must frankly admit that no additional relevant information should derive from this specific analysis. It seems to us that the Referee has the tendency of suggesting analyses that she/he uses to execute in her/his own works.

- The authors correctly point out that the rotational motion of the molecules freezes, maybe it would be a good idea to compute the rotational autocorrelation function, say in the last 50ps? Even better: in different trajectory segments. Again, this is rather standard practice in such studies.

We have performed the analysis of rotational correlation functions that we report below. As the Reviewer can appreciate, the freezing of molecular rotations is gradual, and therefore needs to be assessed at disjoint time windows rather than averaging over the entire trajectories.

In the picture below we report the rotational correlation function $C_{\text{rot}}(t)$ for liquid water in the absence of external EF (cyan) and the $C_{\text{rot}}(t)$ computed at different time windows for the three fields inspected in this work. The first important thing to be noticed is that molecular rotations react differently depending on the strength of the field: during the first 50ps of simulation, molecular rotations are not much affected at 0.05 V/Å, while they noticeably slow down with increasing intensity at 0.10 V/Å and 0.15 V/Å. Upon increasing the simulation time, molecular rotations further slow down and, eventually, they reach complete freezing in the time window [201-250] ps for the cases 0.10 V/Å and 0.15 V/Å, hence further supporting the hypothesis that our f-GW is a glass. We point out here that we are currently working on a detailed analysis of the

(de)coupling of translational and rotational degrees of freedoms and this work will be part of a future publication.

- Error bars are never reported for *any* of the fluctuating microscopic quantities.

The only plot in which error bars should be present but are missing is Fig. 5, for which error bars are reported separately in Fig. 6. Fluctuations of the HBN represent, indeed, a measure of the error associated with the ring statistics. Standard deviation to the mean dipole moments shown in Fig. S2-b of the SI has now been added.

Reviewer #1 (Remarks to the Author):

The authors have done an impressive effort to address all my comments as well as the more extensive comments by reviewer 2. I am satisfied by the modified manuscript and recommend it for publication after the authors have addressed the minor comments below. 1) Page 4, 2nd last line: Please add the intended references in connection with "... as in several previous works." 2) Page 8, 3rd last line: "four-folded configurations" \diamond "four-fold coordinated configurations". 3) Page 9, 4th line: "our sample might be glass." \diamond "our sample might be glassy/might be a glass." As a general personal comment to the authors: The rebuttal was extremely lengthy and with many valuable arguments. I am satisfied that they are reflected sufficiently in the modified manuscript and SI, but I personally find it good practice to put the complete arguments in the manuscript and SI since, evidently, there was some aspect that needed clarification/could be misunderstood.

Reviewer #2 (Remarks to the Author):

See attached (below)

(In the following, my comments from the previous report are in blue, the rebuttals in red)

My main criticism in the first review of this manuscript, is that the novelty aspect is rather weak and does not justify publication in Nature Communications. To me, re-labeling or re-branding an already reported and known finding as “electrofreezing”, is rather an issue of semantics. The facts are simple, the substantial drop in diffusion which the authors report, has already been reported several times with several methods (including ab initio MD).

In their rebuttal, the authors repeat several times that the novel aspect is the longer trajectories. That would be true if the longer trajectories enable them to report new findings (e.g. on longer time scales) that have not been previously reported. But this is not the case, what they report are the same findings, regarding the electrofreezing, RDFs, rotational diffusion, IR spectra, and the topology of the H-Bbond network, with marginal incremental improvements, but no real novelty. Granted, there are minor quantitative differences, and the authors can follow the convergence of the different values along the long trajectory, but at the end of the day, the obtained results do not change the overall picture which has already been reported.

In particular, regarding the electrofreezing of water (the title of the paper), because of the very short time scales accessible to ab initio MD, the authors had to resort to the computed diffusion coefficients to characterize the system as a glass. Using the same metric that the authors have used, other studies have all reported this “electrofreezing”, employing non-polarizable- and polarizable- force fields, as well as ab initio MD simulations. In comparison to the latter, the authors have indeed run longer ab initio MD trajectories, but this in itself – as I detailed in my comments – has not lead to new insights that were absent from the previous works, it just lead to an affirmation of existing findings. Longer MD simulations by themselves do not qualify as a “novel” aspect if the final conclusion is the same. The authors show a new Figure, to prove that in a particular case the diffusion coefficient keeps dropping at longer simulation times, but this does not lead to qualitatively new conclusions because the diffusion coefficient was already that of a solid at short times.

This brings me to my second main criticism, which is regarding the way they cite previous work, specifically, when that work is done by others, and strongly overlaps with their findings. In my previous report I wrote:

“The authors seem to have overlooked some important studies that address exactly the same question, *sometimes with almost exactly the same method and the same field strengths*, and reach more or less the same conclusions. **Sometimes the claim of novelty is explicitly made, and sometimes a finding is reported without due comparison to similar – or identical – findings in already published studies.**”

Sadly, the authors’ rebuttal employs the same strategy in dealing with previous works. So now they do cite some of the previous works, but in a manner that does not show how far those cited works show the same findings. This gives the reader who is not aware of those works, an illusion of novelty that is simply not there. Instead of fixing this issue in the revised manuscript, the issue is actually worse. It is one thing not to be aware of previous similar works, but it is another matter to be aware of it, then only to cite it in a context that does not mention that those studies reports the same findings.

A typical case is their response to a comment from my side regarding the diffusion coefficient, which as already explained, lies at the core of their argument: “The previously obtained values ... indicate a drop of D (diffusion coefficient) by about one order of magnitude at $0.1 - 0.2 \text{ V/\AA}$, in agreement with

the current study.” So I am plainly stating that comparison with previous findings shows that the results with longer trajectories simply match the findings of the previous results.

I quote here part of the authors rebuttal: “As we reported above, the Reviewer compares our results, related to the last 50 ps of long simulations, with longer time averages of often shorter simulations. Therefore, our results are indeed novel. While they may have been present in previous works or not, their presence has been overlooked in the past. “

First of all, what the authors say in their statement is not true. Their results would be novel if the findings were novel. They do not challenge my statement that their results agree with previous findings, they just claim that the same results obtained with longer trajectories, are novel results. I am sorry but this is not novelty, this is just a confirmation of previous findings.

In the revised manuscript, no such comparison to previous results is given at all, not a hint that previous works have reported the same trend in diffusion coefficient with field strength. Going through the whole article, one is lead to believe that this electrofreezing is a novel finding, and that no one has computed these diffusion coefficients before. They do not state in the article what they state in their rebuttal, that the results was obtained before but the trajectories are now longer, they merely do not mention that these results exist (and sorry for repeating myself: they exist at the ab initio MD level but with shorter trajectories). The current manuscript in fact fully supports polarizable force field results by Shafiei et al. which I mentioned in my report, but again, which the authors still totally ignore in the revised manuscript and in the rebuttal. This is a very valuable piece of information that is unfortunately not presented to the reader. If the novelty is absent from the results, then at least such inter-method comparisons would be enlightening to highlight the strengths and limitations of each method. I urged the authors in my first report to bring out such comparisons because they would be useful to the community, spent a lot of time in a genuine attempt to point out to the authors the literature which they overlooked, unfortunately this was ignored (also see below). The authors have chosen instead to explain that now the novel aspect is not entirely in the electrofreezing (the title of the paper), but in the difference between the relaxation time scale of individual water molecules, and the hydrogen bond network, I will comment on this below.

But before I leave this point, I also do not understand how the authors have decided that those previous results were overlooked! Overlooked by whom? Perhaps in those studies, the authors did not state the results in hyperbolic terms, but were rather careful, because, as for example mentioned by Futera et al. all these studies (and the current study), are being done at field values above the limit of dielectric breakdown. Any sub-nanosecond simulation will fail to capture the long-time consequences of this!

“... According to the authors, the most important and novel finding is the electrofreezing of water at a threshold field of 0.1 V/\AA , or rather in particular, that this is observed under ambient conditions, as the authors emphasize twice in the abstract. The transition to the nonlinear response regime is already known to happen at $\sim 0.1 \text{ V/\AA}$ [10.1080/14733140500352322].”

Response:

“The Reviewer cites a manuscript reporting methodologies and techniques completely different from ours. In the cited work [10.1080/14733140500352322], the Author performs classical simulations on alternating electromagnetic fields, whereas we focus on ab initio simulations on continuous electric fields. Nonetheless, words and concepts related to “structural/phase transition”, “electrofreezing” or “glass” never appear in the mentioned paper; not a single statement is present on the structural transitions induced by the electric field. However, we have cited the suggested work (now Ref. [32]) in

the Introduction remarking, however, that it refers to classical molecular dynamics and oscillating fields: [...] based on classical molecular dynamics. According to these studies, liquid water undergoes electrofreezing to crystalline ice when exposed to external static EFs in the order of $\sim 0.5 \text{ V/\AA}$ [30, 31], and the effects of oscillating EFs have also been investigated [32].”

In the course of my first report, I have cited several works, some employing classical non-polarizable or polarizable force fields, and even ab initio MD. At least a couple of those indeed report the freezing at 0.1 V/\AA under static fields and ambient conditions, as the authors would immediately find if they refer to them. It would be quite exhausting to cite every time every single paper that reports similar findings, as this is indeed a heavily studied subject. So for the sake of argument, if the authors do not think that this particular work is relevant, what about the other literature, which – at least now – they are aware of? For example, I mentioned the work from Shafiei et al, which employed static fields, and which the authors totally ignore in their rebuttal, and in the revised manuscript! They cite now the work from Futera et al (ab initio MD, static fields), but only regarding the vibrational spectra, and curiously, in this paragraph of the response, they ignore all this.

Interestingly, now in the current version, the authors refer to previous work as indeed reporting “electrofreezing”! True enough. But now, this new sentence “According to these studies, liquid water undergoes electrofreezing to crystalline ice...”, is problematic as it contradicts the statement by the authors, twice, that “Our work represents the first evidence of electrofreezing of liquid water at ambient conditions”.

According to the authors own claims, perhaps this statement should be modified to say that this is the first evidence using ab initio MD, and static fields? Except that even this is also not true. In my comments, I have pointed them to another study which employed static fields and ab initio MD (the study by Futera et al.), which again reports electrofreezing. And this is what I find troubling. The authors know that I am aware of that study, the authors are surely now aware of that study (It turns out they have cited it in a previous publication). yet, the authors never mention it in the context of electrofreezing! Interestingly, after my criticism, now they cite the study by Futera et al. but not in the context of electrofreezing, but only in the reporting of the IR spectra, as if this is the only thing reported in that work about the effects of static fields on liquid water under ambient conditions using ab initio MD! This is exactly the attitude I strongly criticized in my review. I do not think that this is a good practice.

- The Reviewer cites manuscripts in which, instead, averages are ran over the entire trajectories, which are often too short to capture the transition to f-GW.... In the revised manuscript, we have quantified the difference in looking at the last 50ps with respect to running averages over 250ps. The revised manuscript reports new Figures showing averages over the entire trajectories. While the results fully agree with the data reported in the literature to which the Reviewer is referring, they lead to misleading conclusions and hide the transition to f-GW... Fig. S10 in the SI, clearly show how running long averages leads to wrong conclusions”

The authors are merely ignoring my other comment, where I clearly state that the exact same drop in diffusion coefficient has been previously reported. Assuming the claim is true about equilibration in previous works (which is not true, c.f. the work from Shafeiei et al.), but assuming this, Fig. 10 in the SI actually does not show what the authors claim. Longer runs lead to a better converged estimate of the diffusion coefficient, no one would dispute this. But the important point is, both lines in Fig S10b correspond to a solid! In the authors own data set, whether one averages over the entire trajectory or over the last 50 ps, we reach the same conclusion: we have a solid! No

wrong conclusions reached, just a more accurate diffusion coefficient of the solid is obtained from the last 50 ps, which is not surprising. Again, I am not claiming that longer MD simulations are not delivering better converged results, I am just stating that the same conclusions have been already reached, and I do not know really what the authors are arguing about. I cited a previous work reporting a drop of the same one order of magnitude in the diffusion coefficient, at the same field strength that the authors are using, exactly like the current findings, so what wrong conclusions were reached?

- We have also further emphasised the novelty of our new approach in the revised manuscript. We have added the following sentence to the Abstract: While the response of single water molecules is almost instantaneous, the cooperativity of the hydrogen bonds induces slower reorganizations that can be captured by dividing the trajectories in disjoint time windows and by performing analysis on each of them separately.

Again this is not a new finding, The orientation of a water molecule in the diffusive regime (roughly below field strength of 0.4×10^{-9} V/m, see 10.1038/s41598-019-46449-5), has a timescale of about 1 ps, for example see the works by Alenka Luzar, or Damien Laage. In fact, the time scale from the new Figure S3 is the same as the red trace in Figure 1a in 10.1038/s41598-019-46449-5 which shows the time-development of the average of the distribution under a constant static field, with the ~ 50 ps (probably a little bit less) equilibration time (the work is again not cited in the revised manuscript, because the authors responded that it is not relevant). Hydrogen bond network rearrangements on the other hand are known to be much slower, the static dielectric response of water is known to have slow components (even slower under large static fields) involving cooperative reorganization of a number of water molecules (exact number still disputed). The practice of dividing a time-dependent trajectory into time windows for analysis is a standard practice in non-equilibrium MD simulations, with the span of the time windows being given by the time scale of the relaxation processes.

We emphasise that Ref. [39] (now Ref. [40]) is a work of one of us and refers the reader to the paper of Futera and English [10.1063/1.4994694] (see Ref. [5] of Ref. [40] therein) here mentioned by the Reviewer. Therefore, we do not claim anything new here. A more careful inspection of our manuscript can surely show that the IR spectra is a contextual introductory tile. However, we clearly mention in our manuscript that the IR spectra reported are computed over the last 50ps of very long ab initio molecular dynamics simulations, hence being it itself novel compared to whatever the Reviewer cites:

I was naturally commenting on citations in the current manuscript, not about citations in one of your previous manuscripts. Futera et al. predate your mentioned manuscript by two years, and I think it should be cited, which you now do in the revised manuscript. I apologize if there was a misunderstanding, but I find a consistent trend in the article, to only compare the findings of the current work to previous findings, if the previous findings were done on a different system, but to ignore such comparisons when it comes to previous work done on liquid water under static fields and ambient conditions.

In the revised manuscript, Futera et al. are now cited in in the context of IR spectra (but not RDF, electrofreezing, or rotational ACFs, all of which are reported by Futera et al., to the same end). Regardless of citation issues, the same question remains, what new features do you find in the IR spectra that Futera and English did not find in 2017? I mean, what new features besides the fact that your averages are over the last 50 ps? Futera et al. report only the two peaks where they find a significant change, that is the librational and the O-H stretch peaks, including the new peak at the high-

frequency edge of the librational peak. What they missed to report, and what is new in your findings, is the enhancement of the weak libration+bending band. I think it is good practice to make these things clear in the manuscript. The authors always make statement that “we do not claim anything new here”. Well, why do we cite previous work then? Not citing the same finding reported in previous work, is claiming there is something new, isn't it?

Finally, the Reviewer claims that the results reported in [10.1038/s41598-019-46449-5] are relevant to our work. The authors of the cited work use a pulsed squared electric field (of duration equal to 500 fs) whilst we apply a constant electric field. Therefore, such comparison is not justified. Our paper shows exactly the opposite, in that pulsed electric fields cannot lead to

The authors need to have a closer look at that paper. The paper does not only use a square pulse (though this is the focus), results with other continuous static fields are also reported. As I already mentioned, the “novel” finding regarding the time-scale of single molecule orientation is reported there in the first Figure. Here, I was talking about a very specific concept, which is the effect of intense static fields on H-bond strengths. Again, please have a look at the analysis of the H-bonds under the square pulse in that work, it will be more clear what I exactly meant. As I mentioned numerous times in the previous report, giving the relevant references, roughly speaking it is well-known that oscillating fields induce disorder, and static-fields induce order, in liquid water.

We stress here the fact that the cited work [10.1038/s41598-019-46449-5] never mentions – in any single statement – the possibility of freezing liquid water, the essence of our work.

I never claimed that this particular work mentions electrofreezing, it was cited in a different context as my original comment clearly shows. Other papers however do report the freezing as I already mentioned in my previous comments, and there the authors remain silent about that fact!

Here too, we are not claiming any novelty in the alignment of water molecules with the direction of the field. We would like to mention that one of us has published several papers reporting this observation (see, e.g., the most recent one being JCP 158, 184502 2023). The Reviewer is missing the main message and the broader picture not previously reported, which is new: the alignment of single molecules is extremely fast, but the HBN requires time, in a fully dynamical process, to adjust. Based on this idea, we probe different time windows and uncover the electrofreezing to f-GW. This concept is novel and never adopted before. We emphasise it several times in the original version of the manuscript. In the Introduction:

In the first comment in the rebuttal, the authors mention that they cite 75 references, well, none of those is cited regarding the observed alignment, only a self-citation (in discussing IR spectrum) from 2019. Again, I am puzzled by claims of the authors that “they are not claiming any novelty”, while they report their findings without citing the same findings reported previously! The fact that one of the authors published something similar this year, 2023, does not excuse the authors from not citing *any* previous works from several years ago.

Again, regarding the fact that single molecule alignment is fast, but collective alignment is slow, as previously mentioned, this is by far not a new message. The so-called single molecule fast alignment (few tens of picoseconds) is exactly reported in 10.1038/s41598-019-46449-5 Fig. 1A (using a static field, not a square pulse as the authors in a previous comment claimed). The authors in a previous comment also mentioned that the aforementioned work is irrelevant to the current publication, despite

now it apparently reporting one of the “novel” findings. As regards to the electrofreezing, and its novelty, I have already addressed this in a previous comment.

The main message, reported in this paragraph, was absent from the title and the abstract of the previous version of the manuscript. It is still absent from the title which only reports on electrofreezing of water.

I am skipping comment on several other points, because there is not point repeating myself and the first few paragraphs of this report summarize the matter. Maybe just a new point is raised here:

We do not use static dipoles. Instead, dipole moments distributions (Fig. S2) are determined directly from the electronic structure and, in particular, from the centres of the Maximally Localised Wannier Functions. Thus, dynamical variations of the dipoles are fully considered from the underlying electron density. This is now explicitly stated in the caption of Fig. S2.

This is only a side-point, but I think the authors do not understand the term “static dipoles” vs. dynamic dipoles. It does not mean fixed dipoles like e.g. force fields. In this context, the choice of the authors indeed corresponds to static dipoles, please refer to Pasquarello et al. 10.1103/PhysRevB.68.174302.

Reviewer #1

Query: “The authors have done an impressive effort to address all my comments as well as the more extensive comments by reviewer 2. I am satisfied by the modified manuscript and recommend it for publication after the authors have addressed the minor comments below.”

Authors’ reply: We are grateful to the Reviewer for the very positive overall comments and for recommending our work for publication.

Query: “Page 4, 2nd last line: Please add the intended references in connection with “... as in several previous works.”

Authors’ reply: We thank the Reviewer for this hint. We have added the references in the revised manuscript.

Query: “Page 8, 3 last line: “four-folded configurations” → “four-fold coordinated configurations”.”

Authors’ reply: We have fixed the typo.

Query: “Page 9, 4 line: “our sample might be glass.” → “our sample might be glassy/might be a glass.””

Authors’ reply: We have modified the sentence accordingly.

Query: “As a general personal comment to the authors: The rebuttal was extremely lengthy and with many valuable arguments. I am satisfied that they are reflected sufficiently in the modified manuscript and SI, but I personally find it good practice to put the complete arguments in the manuscript and SI since, evidently, there was some aspect that needed clarification/could be misunderstood.”

Authors’ reply: We thank the Referee for this personal comment, on which we agree. We have added the most relevant findings reported and commented in our previous rebuttal letter in the main text and in the SI file to make these data available and the respective aspects clearer to the readership (see the changes when presenting the results in connection with the existing literature (pages from 2 to 7), the new Fig. S11 of the SI, and related discussion). We have introduced the following clarifications that will clear possible misunderstandings.

We have clarified and justified the adoption of time windows of 50ps in the revised Introduction:

The length of the observation time windows is crucial to capture the correct physics: too long-time windows wash out the information on the non-ergodicity, an issue particularly relevant when the system is drifting out of equilibrium. Too small-time windows, on the other hand, sample a limited portion of the configurational space inducing artificial suppression of the first moments of the quantities of interest. We found that, for our system, time windows of 50 ps provide the right balance between the out of equilibrium nature and the physical behaviour of fluctuations.

We have introduced in the Results section a detailed discussion on the difference between our data and the data reported in the literature which, we posit, are affected by the averaging over long time scales and by too short simulation times:

The corresponding diffusion coefficients at the strength of the fields here studied are reported in the inset. Using the diffusivity for the case with no external field as a baseline, we observe a reduction of the diffusion coefficient up to 12.5 times in the case of $0.15 \text{ V}/\text{\AA}$. Such a substantial drop is several times larger than the field-induced drops reported elsewhere and for similar fields. In particular, Ref. [42] reports a drop of 1.9 times induced by a field of $0.10 \text{ V}/\text{\AA}$ from AIMD simulations. The reduced drop in diffusivity reported in Ref. [42] is the result of the short timescales there considered, i.e., 100 ps while, as we have mentioned, electrofreezing occurs after $\sim 150 - 200$ ps. We observe a comparable drop for the case of $0.10 \text{ V}/\text{\AA}$ if, like in Ref. [42], we limit ourselves to the first 100 ps of simulations. Ref. [48] investigates several classical potentials and indicates that a field of $0.15 \text{ V}/\text{\AA}$ induces a drop between 1.2 times to 2.5 times depending on the potential. According to Ref. [48], a more substantial drop of ~ 5.5 times occurs when the EF is increased to $0.20 \text{ V}/\text{\AA}$, conditions that induce nonnegligible water ionization in our extended AIMD simulations. Therefore, we posit that our enhanced reduction is the effect of windowing the time series on timescales that allow for electrofreezing to occur.

In the Results section, we also now report the results on the freezing of rotational degrees of freedom, and we report the corresponding plot in the Supporting Information.

We report in Fig. S11 the profiles of the rotational autocorrelation functions computed at consecutive time windows for all the cases here investigated and compared against the case without external field. The gradual reduction of rotational degrees of freedom with time, induced by the presence of EFs is clearly visible; eventually, rotational degrees of freedom are fully frozen within the last 50 ps for EFs of $0.10 \text{ V}/\text{\AA}$ and $0.15 \text{ V}/\text{\AA}$. Like for the diffusivity, our results on the rotational dynamics in the early stages of the simulations (100 ps) are qualitatively in agreement with Ref. [42]. Therefore, the freezing of translational and rotational degrees of freedom occurring after $\sim 150 - 200$ ps is strongly indicative that we are witnessing a dynamic arrest and a transition to f-GW.

In order to fully contextualize our work, we have added a remark in the Results section on the importance of running averages over a properly chosen time windows:

We emphasize here the importance of choosing meaningful observation windows: too long time windows imply running averages on states too far from each other, while too short time windows imply incorrectly sampling the configurational space. For instance, time windows of 20 ps would show an artificial suppression of the fluctuations of the HBN topology even at earlier times, when the system is instead in the liquid state.

We have added in the Supporting Information the following discussion on the behaviour of the rotational autocorrelation functions:

Finally, to estimate the effects of EFs on the rotational dynamics, we have also computed the rotational autocorrelation functions $C_{\text{rot}}(t)$, that we report in Fig. S11 for the three fields and at consecutive time windows. Fig. S11 also reports the profile of $C_{\text{rot}}(t)$ computed for liquid water in the absence of EFs. The lowest field here inspected (upper panel) affects the rotational degrees of freedom mostly only after ~ 50 ps. The profile of $C_{\text{rot}}(t)$ at higher times indicates that rotational degrees of freedom become increasingly slow but are always active, although very sluggish after ~ 200 ps. Upon increasing the field strength to $0.10 \text{ V}/\text{\AA}$ and to $0.15 \text{ V}/\text{\AA}$ (middle and lower panel, respectively), molecular rotations become rapidly very slow even within the first 50 ps, this effect being more pronounced at the highest field. For both cases, the fields induce a complete freezing of molecular rotations after $150 - 200$ ps, as captured by the plateau of $C_{\text{rot}}(t)$.

Reviewer #2

Query 1: “My main criticism in the first review of this manuscript, is that the novelty aspect is rather weak and does not justify publication in Nature Communications. To me, re-labeling or re-branding an already reported and known finding as “electrofreesing”, is rather an issue of semantics. The facts are simple, the substantial drop in diffusion which the authors report, has already been reported several times with several methods (including ab initio MD).”

Authors’ reply: We thank the Reviewer for the possibility to make this aspect clearer since, in our opinion, this is not merely a semantic issue, and we are not re-labeling nor re-branding already reported results. We decided to take this occasion to clarify, via quantitative data, the inaccuracy of the Reviewer’s statement “*the substantial drop in diffusion which the authors report, has already been reported several times with several methods (including ab initio MD)*”. We summarize in the following Table the diffusion coefficients reported in the literature cited by the Reviewer along with our results. The diffusion coefficients are reported in Å²/ps. Diffusion coefficients are not reported in *English, Mol. Phys. 104, 243-253 (2006)* or in *Elgabarty, Kaliannan & Kühne, Sci. Rep. 9, 10002 (2019)*.

Futera & English, JCP 147, 031102 (2017), Table II	Shafiei, von Domaros, Bratko & Luzar, JCP 150, 074505 (2019), Figure S3 of the SI	Our work
D(No field) = 0.155 D(0.05 V/Å) = 0.111 D(0.10 V/Å) = 0.081	SPC/E : D(No field)≈0.26 D(0.15V/Å)≈0.22	D(No field)=0.25 D(0.15 V/Å)=0.02
	BK3 : D(No field)≈0.20 D(0.15V/Å)≈0.08	
	SWM4-NDP : D(No field)≈0.26 D(0.15 V/Å)≈0.17	

We would like to expand on these results.

In Futera & English, first column, the field 0.10 V/Å **does not even halve the diffusion coefficient** obtained in the absence of any field (i.e., D(0.10 V/Å) is only **1.9 times** smaller than its field-free counterpart). We notice here that D(No field) itself already corresponds to a very sluggish dynamics. In Shafiei, von Domaros, Bratko & Luzar, second column, the authors report diffusion coefficients for three classical potentials. For the SPC/E model, they observe a reduction in diffusion coefficient from the case of no field to the case of 0.15 V/Å by only **1.2 times**. For the BK3 model the reduction is only **2.5 times**, for SWM4-NDP the reduction is of only **1.5 times**. We notice that, for the case of BK3, the authors report a drop in diffusion coefficient from the case of no field to the case of 0.20 V/Å by 5.5 times, but AIMD simulations at this field strength indicates that non-negligible water ionizations occur.

Our work reports a drop in diffusion coefficient by a factor of **12.5 times**, i.e., **one order of magnitude higher** than in the listed literature. Therefore, the Reviewer’s claim “*The facts are simple, the substantial drop in diffusion which the authors report, has already been reported several times with several methods (including ab initio MD)*.” is simply not accurate. Moreover, the Reviewer should be well aware that Futera & English performed simulations of 100 ps while we perform much longer simulations, from 250ps to up to 500ps. To clarify the discrepancy with

the drop in diffusion between us and Futera & English, we have computed the diffusion coefficient for the case of 0.10 V/\AA and for the first 100ps only. Our results show a diffusion coefficient of $0.097 \text{ \AA}^2/\text{ps}$, corresponding to a drop of 2.6 times with respect to the zero-field case, comparable to that reported in Futera & English on the same timescales.

We have added all these considerations in the revised manuscript:

The corresponding diffusion coefficients at the strength of the fields here studied are reported in the inset. Using the diffusivity for the case with no external field as a baseline, we observe a reduction of the diffusion coefficient up to 12.5 times in the case of 0.15 V/\AA . Such a substantial drop is several times larger than the field-induced drops reported elsewhere and for similar fields. In particular, Ref. [42] reports a drop of 1.9 times induced by a field of 0.10 V/\AA from AIMD simulations. The reduced drop in diffusivity reported in Ref. [42] is the result of the short timescales there considered, i.e., 100 ps while, as we have mentioned, electrofreezing occurs after $\sim 150 - 200$ ps. We observe a comparable drop for the case of 0.10 V/\AA if, like in Ref. [42], we limit ourselves to the first 100 ps of simulations. Ref. [48] investigates several classical potentials and indicates that a field of 0.15 V/\AA induces a drop between 1.2 times to 2.5 times depending on the potential. According to Ref. [48], a more substantial drop of ~ 5.5 times occurs when the EF is increased to 0.20 V/\AA , conditions that induce nonnegligible water ionization in our extended AIMD simulations. Therefore, we posit that our enhanced reduction is the effect of windowing the time series on timescales that allow for electrofreezing to occur.

We emphasize here that a drop in the diffusion coefficient, often evoked by the Reviewer in her/his reply, does not constitute a proof of a transition to a glass, nor of any transition for that matter. We also remark again, as we have already done in the previous iteration and in our previous response, that the drop in the network *dynamics* is what signals a transition to a glass. We have added the following considerations in the revised manuscript:

We report in Fig. S11 the profiles of the rotational autocorrelation functions computed at consecutive time windows for all the cases here investigated and compared against the case without external field. The gradual reduction of rotational degrees of freedom with time, induced by the presence of EFs is clearly visible; eventually, rotational degrees of freedom are fully frozen within the last 50 ps for EFs of 0.10 V/\AA and 0.15 V/\AA . Like for the diffusivity, our results on the rotational dynamics in the early stages of the simulations (100 ps) are qualitatively in agreement with Ref. [42]. Therefore, the freezing of translational and rotational degrees of freedom occurring after $\sim 150 - 200$ ps is strongly indicative that we are witnessing a dynamic arrest and a transition to f-GW.

and the following considerations in the Supplementary Information:

Finally, to estimate the effects of EFs on the rotational dynamics, we have also computed the rotational autocorrelation functions $C_{\text{rot}}(t)$, that we report in Fig. S11 for the three fields and at consecutive time windows. Fig. S11 also reports the profile of $C_{\text{rot}}(t)$ computed for liquid water in the absence of EFs. The lowest field here inspected (upper panel) affects the rotational degrees of freedom mostly only after ~ 50 ps. The profile of $C_{\text{rot}}(t)$ at higher times indicates that rotational degrees of freedom become increasingly slow but are always active, although very sluggish after ~ 200 ps. Upon increasing the field strength to 0.10 V/\AA and to 0.15 V/\AA (middle and lower panel, respectively), molecular rotations become rapidly very slow even within the first 50 ps, this effect being more pronounced at the highest field. For both cases, the fields induce a complete freezing of molecular rotations after $150 - 200$ ps, as captured by the plateau of $C_{\text{rot}}(t)$.

Finally, here we would like to re-iterate again that **none of the articles listed above report even the slightest allusion to the observation of a structural transition, of electrofreezing, or of a liquid-glass transition, the**

latter being the main topic of the current work. Considering all these qualitative and quantitative considerations, we are not simply “re-labeling” nor “re-branding” anything: our results are, indeed, novel, as also acknowledged by Reviewer #1.

Query 2,3: “In their rebuttal, the authors repeat several times that the novel aspect is the longer trajectories. That would be true if the longer trajectories enable them to report new findings (e.g. on longer time scales) that have not been previously reported. But this is not the case, what they report are the same findings, regarding the electrofreezing, RDFs, rotational diffusion, IR spectra, and the topology of the H-Bond network, with marginal incremental improvements, but no real novelty. Granted, there are minor quantitative differences, and the authors can follow the convergence of the different values along the long trajectory, but at the end of the day, the obtained results do not change the overall picture which has already been reported.

In particular, regarding the electrofreezing of water (the title of the paper), because of the very short time scales accessible to ab initio MD, the authors had to resort to the computed diffusion coefficients to characterize the system as a glass. Using the same metric that the authors have used, other studies have all reported this “electrofreezing”, employing non-polarizable- and polarizable- force fields, as well as ab initio MD simulations. In comparison to the latter, the authors have indeed run longer ab initio MD trajectories, but this in itself – as I detailed in my comments – has not lead to new insights that were absent from the previous works, it just lead to an affirmation of existing findings. Longer MD simulations by themselves do not qualify as a “novel” aspect if the final conclusion is the same. The authors show a new Figure, to prove that in a particular case the diffusion coefficient keeps dropping at longer simulation times, but this does not lead to qualitatively new conclusions because the diffusion coefficient was already that of a solid at short times.”

Authors’ reply: We respectfully disagree with the Reviewer on several aspects. We do not claim that longer trajectories are the novel aspect. As we have reported in the previous iteration, the novel aspect is the detailed analysis of the hydrogen bond network and the suppression of its fluctuations. These analyses prove that AIMD simulations longer than ≈ 150 - 200 ps are necessary to observe the transition. Simulations of this length and explicitly accounting for the crucial electronic polarization effects were never reported before, another fully novel aspect featuring our work. As we have also reported above, the reduction of the diffusion is not a proof of transition to a glass, as the Reviewer seems to imply. Nonetheless, the reduction reported in the suggested references is minimal and far from being informative compared to ours (see Table at page 3). Our conclusions are, therefore, not just qualitative but, rather, quantitative.

Query 4, 5, 6: “This brings me to my second main criticism, which is regarding the way they cite previous work, specifically, when that work is done by others, and strongly overlaps with their findings. In my previous report I wrote:

“The authors seem to have overlooked some important studies that address exactly the same question, sometimes with almost exactly the same method and the same field strengths, and reach more or less the same conclusions. Sometimes the claim of novelty is explicitly made, and sometimes a finding is reported without due comparison to similar – or identical – findings in already published studies.”

Sadly, the authors’ rebuttal employs the same strategy in dealing with previous works. So now they do cite some of the previous works, but in a manner that does not show how far those cited works show the same findings. This gives the reader who is not aware of those works, an illusion of novelty that is simply not there. Instead of fixing this issue in the revised manuscript, the issue is actually worse. It is one thing not to be aware of previous similar

works, but it is another matter to be aware of it, then only to cite it in a context that does not mention that those studies reports the same findings.”

Authors’ reply: We thank the Reviewer for giving us the opportunity to better contextualize some of our results with those present in the literature. We have added several sentences and extensive discussions in the revised version of the article, with extended and direct quantitative comparisons of our data with those available from the literature are now included (see pages 2-7).

Query 7: “A typical case is their response to a comment from my side regarding the diffusion coefficient, which as already explained, lies at the core of their argument: “The previously obtained values ... indicate a drop of D (diffusion coefficient) by about one order of magnitude at 0.1 – 0.2 V/Å, in agreement with the current study.” So I am plainly stating that comparison with previous findings shows that the results with longer trajectories simply match the findings of the previous results.”

Authors’ reply: We redirect the Reviewer to our previous answers and to the revised manuscript. The drop in D in the references suggested by the Reviewer are one order of magnitude *smaller* than ours.

Query 8, 9: “I quote here part of the authors rebuttal: “*As we reported above, the Reviewer compares our results, related to the last 50 ps of long simulations, with longer time averages of often shorter simulations. Therefore, our results are indeed novel. While they may have been present in previous works or not, their presence has been overlooked in the past.* “

First of all, what the authors say in their statement is not true. Their results would be novel if the findings were novel. They do not challenge my statement that their results agree with previous findings, they just claim that the same results obtained with longer trajectories, are novel results.

I am sorry but this is not novelty, this is just a confirmation of previous findings.”

Authors’ reply: As detailed in the reply to **Query 1**, our results are novel, and they do not represent a sterile confirmation of previous findings. The fact that in the literature a partial slowing down of the water dynamics under static fields was observed does not mean – *in any way* – that other people reported on the electrofreezing of liquid water at ambient conditions. To prove the latter finding, indeed, several other properties – including, e.g., the fluctuation of the H-bond network, the ring statistics, and the evaluation of the potential energy, just to cite a few of the wide-spectrum analyses we report for the first time in this context – have to be carefully taken into account in very long AIMD simulations, explicitly handling strong electronic polarization effects. All these aspects are novel and have never been reported in the literature. Thus, our findings are novel (please refer also to our reply to **Query 1** and others).

Query 10: “In the revised manuscript, no such comparison to previous results is given at all, not a hint that previous works have reported the same trend in diffusion coefficient with field strength. Going through the whole article, one is lead to believe that this electrofreezing is a novel finding, and that no one has computed these diffusion coefficients before. They do not state in the article what they state in their rebuttal, that the results was obtained before but the trajectories are now longer, they merely do not mention that these results exist (and sorry for repeating myself: they exist at the ab initio MD level but with shorter trajectories). Indeed, the current manuscript fully supports polarizable force field results by Shafiei et al. which I mentioned in my report, but again, which the authors still totally ignore in the revised manuscript and in the rebuttal. This is a very valuable piece of information that is unfortunately not presented to the reader. If the novelty is absent from the results, then at least such inter-method comparisons would be enlightening to highlight the strengths and limitations of each method. I urged the authors in my first report to bring out such comparisons because they would be useful to the community, spent a

lot of time in a genuine attempt to point out to the authors the literature which they overlooked, unfortunately this was ignored (also see below). The authors have chosen instead to explain that now the novel aspect is not entirely in the electrofreezing (the title of the paper), but in the difference between the relaxation time scale of individual water molecules, and the hydrogen bond network, I will comment on this below.”

Authors’ Reply: We invite the Reviewer to re-read the papers she/he is mentioning. In those papers, there is no mention about electrofreezing because the authors simply don’t observe it. Their diffusion coefficients are either not reported ([*Mol. Phys.* **104**, 243-253 (2006); *Sci. Rep.* **9**, 10002 (2019)]), too high or with a too weak drop ([*JCP* **147**, 031102 (2017); *JCP* **150**, 074505 (2019)]). The revised manuscript reports an extensive discussion on the data available from the suggested literature and how they compare with ours, see our reply to **Query 1**.

Query 11: “But before I leave this point, I also do not understand how the authors have decided that those previous results were overlooked! Overlooked by whom? Perhaps in those studies, the authors did not state the results in hyperbolic terms, but were rather careful, because, as for example mentioned by Futera et al. all these studies (and the current study), are being done at field values above the limit of dielectric breakdown. Any sub-nanosecond simulation will fail to capture the long-time consequences of this!”

Authors’ reply: The strength of the EF necessary for achieving the dielectric breakdown in water is a delicate and still open question. As also explicitly stated by Luzar and co-workers [*JCP* **150**, 074505 (2019)], although the commonly accepted dielectric strength of bulk water is about 0.007 V/\AA , significantly higher thresholds are indicated by experiments in μm films of deionized water [*Rev. Sci. Instrum.* **81**, 054702 (2010)]. Moreover, fields up to $\approx 0.01\text{-}0.1 \text{ V/\AA}$ are known to be present and pervade water in ion channels [*JCP* **122**, 234706 (2005)] and in the proximity of colloids [*PNAS* **95**, 15169 (1998)] or liquid crystals [*CPL* **128**, 449 (1986)]. Furthermore, it is nowadays well-known that local electric fields larger than $1\text{-}2 \text{ V/\AA}$ are continuously present in water and aqueous solutions, as re-iterated in many papers of the English group [*JCP* **147**, 031102 (2017); *JCP* **119**, 11806-11813 (2003); *J. Mol. Liq.* **389**, 122675 (2023)], recently quantified by some of us [*JPCL* **14**, 7808-7813 (2023)], by Kathman and co-workers [*JPCB* **117**, 10869-10882 (2013)], by Garofalini and co-workers [*JPCB* **127**, 3392-3401 (2023)], and many others. Finally, as shown in the SI of Ref. [*PRL* **108**, 207801 (2012)] (Fig. 1), the electronic polarization of liquid water treated at standard GGA Density Functional Theory (DFT) levels follows a linear regime at electric fields whose intensity exceeds of more than one order of magnitude (i.e., $> 1 \text{ V/\AA}$) those we have applied in the current work ($\approx 0.1 \text{ V/\AA}$). At these very extreme field conditions ($> 1 \text{ V/\AA}$), the band-gap of liquid water has been reported to be larger than 3.5 eV [*PRL* **108**, 207801 (2012)], indicating that metallization does not occur.

Query 14: “In the course of my first report, I have cited several works, some employing classical non-polarizable or polarizable force fields, and even ab initio MD. At least a couple of those indeed report the freezing at 0.1 V/\AA under static fields and ambient conditions, as the authors would immediately find if they refer to them. It would be quite exhausting to cite every time every single paper that reports similar findings, as this is indeed a heavily studied subject. So for the sake of argument, if the authors do not think that this particular work is relevant, what about the other literature, which – at least now – they are aware of? For example, I mentioned the work from Shafiei et al, which employed static fields, and which the authors totally ignore in their rebuttal, and in the revised manuscript! They cite now the work from Futera et al (ab initio MD, static fields), but only regarding the vibrational spectra, and curiously, in this paragraph of the response, they ignore all this.”

Authors’ reply: We thank the Reviewer for the possibility of expanding our work also with reference to the existing literature. In the revised version of the manuscript, we have thoroughly highlighted similarities and differences between our findings and the relevant ones present in the literature (see pages 2-7).

Query 15: “Interestingly, now in the current version, the authors refer to previous work as indeed reporting “electrofreezing”! True enough. But now, this new sentence “According to these studies, liquid water undergoes electrofreezing to crystalline ice...”, is problematic as it contradicts the statement by the authors, twice, that “Our work represents the first evidence of electrofreezing of liquid water at ambient conditions.”

Authors’ reply: Such a sentence was present also in the original version of our article and it transparently refers to the well-known two works of Svishchev & Kusalik [*PRL* **73**, 975-978 (1994); *JACS* **118**, 649-654 (1996)], as correctly referenced in the Introduction of our manuscript. These articles nowadays deserve a historical importance; their conclusions refer to classical force-fields molecular dynamics simulations conducted in the supercooled regime and under a single electric field strength equal to 0.5 V/Å. However, a large body of more recent *ab initio* and experimental investigations have shown that already about a half of the latter field intensity (0.3 V/Å) is capable of triggering molecular ionizations and correlated proton transfers [*PRL* **108**, 207801 (2012); *JPCL* **11**, 8983-8988 (2020); *CPL* **519-520**, 1-17 (2012); *APL* **101**, 243110 (2012)], as also duly reported in our work starting the original version (please refer to the Introduction section, page 2).

Query 17: “According to the authors own claims, perhaps this statement should be modified to say that this is the first evidence using *ab initio* MD, and static fields? Except that even this is also not true. In my comments, I have pointed them to another study which employed static fields and *ab initio* MD (the study by Futera et al.), which again reports electrofreezing. And this is what I find troubling. The authors know that I am aware of that study, the authors are surely now aware of that study (It turns out they have cited it in a previous publication). yet, the authors never mention it in the context of electrofreezing! Interestingly, after my criticism, now they cite the study by Futera et al. but not in the context of electrofreezing, but only in the reporting of the IR spectra, as if this is the only thing reported in that work about the effects of static fields on liquid water under ambient conditions using *ab initio* MD! This is exactly the attitude I strongly criticized in my review. I do not think that this is a good practice.”

Autors’ Reply: We are afraid we cannot accept to be accused of bad practice. We have already explained that the work by Futera and English [*JCP* **147**, 031102 (2017)] does not mention electrofreezing, although the Reviewer keeps mentioning the contrary. We invite the Reviewer to re-read the work by Futera and English, and she/he will realize that it doesn’t mention electrofreezing, because electrofreezing has not been observed there. Futera and English report a slight reduction in the diffusion, a reduction which is one order of magnitude smaller compared to ours and is characteristic of a liquid, not of a solid (see Table at page 3 of the current letter). We have extensively discussed this point also in the revised manuscript. The reason for which Futera and English do not observe electrofreezing is simple: they run simulations of only 100ps, while we show that electrofreezing occurs after 150-200ps. Please refer to our reply to **Query 1**.

Query 19: “The authors are merely ignoring my other comment, where I clearly state that the exact same drop in diffusion coefficient has been previously reported. Assuming the claim is true about equilibration in previous works (which is not true, c.f. the work from Shafeiei et al.), but assuming this, Fig. 10 in the SI actually does not show what the authors claim. Longer runs lead to a better converged estimate of the diffusion coefficient, no one would dispute this. But the important point is, both lines in Fig S10b correspond to a solid! In the authors own data set, whether one averages over the entire trajectory or over the last 50 ps, we reach the same conclusion: we have a solid! No wrong conclusions reached, just a more accurate diffusion coefficient of the solid is obtained from the last 50 ps, which is not surprising. Again, I am not claiming that longer MD simulations are not delivering better converged results, I am just stating that the same conclusions have been already reached, and I do not know really what the authors are arguing about. I cited a previous work reporting a drop of the same one order of

magnitude in the diffusion coefficient, at the same field strength that the authors are using, exactly like the current findings, so what wrong conclusions were reached?”

Authors' Reply: We have already replied to the point on the diffusion coefficient. We want just to reiterate that the Reviewer claim *“I clearly state that the exact same drop in diffusion coefficient has been previously reported”* is wrong, as reported in the Table at page 3 of this document, and extensively discussed in the revised manuscript. Finally, Fig S10b clearly shows that the mean square displacement (MSD) computed over the entire trajectory is that of a liquid, not of a solid, which is exactly our point and answers the question about what wrong conclusions were reached.

Query 21: “Again this is not a new finding, The orientation of a water molecule in the diffusive regime (roughly below field strength of $0.4E-9$ V/m, see 10.1038/s41598-019-46449-5), has a timescale of about 1 ps, for example see the works by Alenka Luzar, or Damien Laage. The time scale from the new Figure S3 is the same as the red trace in Figure 1a in 10.1038/s41598-019-46449-5 which shows the time-development of the average of the distribution under a constant static field, with the ~ 50 ps (probably a little bit less) equilibration time (the work is again not cited in the revised manuscript, because the authors responded that it is not relevant). Hydrogen bond network rearrangements on the other hand are known to be much slower, the static dielectric response of water is known to have slow components (even slower under large static fields) involving cooperative reorganization of a number of water molecules (exact number still disputed). The practice of dividing a time-dependent trajectory into time windows for analysis is a standard practice in non-equilibrium MD simulations, with the span of the time windows being given by the time scale of the relaxation processes.”

Authors' Reply: *“The practice of dividing a time-dependent trajectory into time windows for analysis is a standard practice in non-equilibrium MD simulations, with the span of the time windows being given by the time scale of the relaxation processes”*, but has not been adopted in the references suggested by the Reviewer...

Query 23: “I was naturally commenting on citations in the current manuscript, not about citations in one of your previous manuscripts. Futera et al. predate your mentioned manuscript by two years, and I think it should be cited, which you now do in the revised manuscript. I apologize if there was a misunderstanding, but I find a consistent trend in the article, to only compare the findings of the current work to previous findings, if the previous findings were done on a different system, but to ignore such comparisons when it comes to previous work done on liquid water under static fields and ambient conditions.”

Authors' Reply: All relevant articles have been cited and all relevant (and correct) comparisons have been done in the revised manuscript.

Query 24: “In the revised manuscript, Futera et al. are now cited in in the context of IR spectra (but not RDF, electrofreezing, or rotational ACFs, all of which are reported by Futera et al., to the same end). Regardless of citation issues, the same question remains, what new features do you find in the IR spectra that Futera and English did not find in 2017? I mean, what new features besides the fact that your averages are over the last 50 ps? Futera et al. report only the two peaks where they find a significant change, that is the librational and the O-H stretch peaks, including the new peak at the high- frequency edge of the librational peak. What they missed to report, and what is new in your findings, is the enhancement of the weak libration+bending band. I think it is good practice to make these things clear in the manuscript. The authors always make statement that “we do not claim anything new here”. Well, why do we cite previous work then? Not citing the same finding reported in previous work, is claiming there is something new, isn't it?”

Authors' Reply: We have extensively reported the difference between these spectra in the manuscript and in our previous response, as well as in a previous work of one of us [*PCCP* **21**, 21205 (2019)] as already discussed during the previous iteration. In the revised manuscript, we further extend our comparison to rotational ACFs and RDFs. Nonetheless, it should be clear to the Reviewer, by now, that Futera and English refer to much shorter simulation times of 100ps while electrofreezing occurs only after 150-200ps, and that running averages over entire trajectories is incorrect. As a matter of fact, the ACFs reported in Futera & English correspond to those of a liquid, as we now clearly state in the revised manuscript. We have added the following statements in the revised manuscript:

The corresponding diffusion coefficients at the strength of the fields here studied are reported in the inset. Using the diffusivity for the case with no external field as a baseline, we observe a reduction of the diffusion coefficient up to 12.5 times in the case of 0.15 V/Å. Such a substantial drop is several times larger than the field-induced drops reported elsewhere and for similar fields. In particular, Ref. [42] reports a drop of 1.9 times induced by a field of 0.10 V/Å from AIMD simulations. The reduced drop in diffusivity reported in Ref. [42] is the result of the short timescales there considered, i.e., 100 ps while, as we have mentioned, electrofreezing occurs after ~ 150 – 200 ps. We observe a comparable drop for the case of 0.10 V/Å if, like in Ref. [42], we limit ourselves to the first 100 ps of simulations. Ref. [48] investigates several classical potentials and indicates that a field of 0.15 V/Å induces a drop between 1.2 times to 2.5 times depending on the potential. According to Ref. [48], a more substantial drop of ~ 5.5 times occurs when the EF is increased to 0.20 V/Å, conditions that induce nonnegligible water ionization in our extended AIMD simulations. Therefore, we posit that our enhanced reduction is the effect of windowing the time series on timescales that allow for electrofreezing to occur.

Our rotational ACFs, which report the gradual transition to a frozen dynamics and to f-GW, are now reported in the revised version of the SI (see Fig. S11):

We report in Fig. S11 the profiles of the rotational autocorrelation functions computed at consecutive time windows for all the cases here investigated and compared against the case without external field. The gradual reduction of rotational degrees of freedom with time, induced by the presence of EFs is clearly visible; eventually, rotational degrees of freedom are fully frozen within the last 50 ps for EFs of 0.10 V/Å and 0.15 V/Å. Therefore, the absence of translational and rotational degrees of freedom is strongly indicative that we are witnessing a dynamical arrest.

Nonetheless, as reported in our reply to the **Query 1** and in the revised manuscript, we do qualitatively capture the results reported in Futera and English when we limit ourselves to the same shorter time scale of 100ps. Please refer to our answer to **Query 1**.

Query 25: “The authors need to have a closer look at that paper. The paper does not only use a square pulse (though this is the focus), results with other continuous static fields are also reported. As I already mentioned, the “novel” finding regarding the time-scale of single molecule orientation is reported there in the first Figure. Here, I was talking about a very specific concept, which is the effect of intense static fields on H-bond strengths. Again, please have a look at the analysis of the H-bonds under the square pulse in that work, it will be more clear what I exactly meant. As I mentioned numerous times in the previous report, giving the relevant references, roughly speaking it is well-known that oscillating fields induce disorder, and static-fields induce order, in liquid water.

Authors' reply: We thank the Referee for the opportunity to include the current work (i.e., [*Sci. Rep.* **9**, 10002 (2019)]) in the bibliography of our article. Reference to this work has now been made both in the Introduction section (page 2) and when commenting the typical field-induced dipolar re-orientations in the revised manuscript (page 3).

Query 30: “Again, regarding the fact that single molecule alignment is fast, but collective alignment is slow, as previously mentioned, this is by far not a new message. The so-called single molecule fast alignment (few tens of picoseconds) is exactly reported in 10.1038/s41598-019-46449-5 Fig. 1A (using a static field, not a square pulse as the authors in a previous comment claimed). The authors in a previous comment also mentioned that the aforementioned work is irrelevant to the current publication, despite now it apparently reporting one of the “novel” findings.””

Authors’ reply: We claim as the suppression of the network dynamics. Nonetheless, we have included and cited the mentioned paper (i.e., [*Sci. Rep.* **9**, 10002 (2019)]) in the Introduction and in the Results sections along with other papers published by some of us, by English and co-workers, and by Luzar and co-workers that all report on the water dipolar re-orientations under the electric field influence.

Query 34: “This is only a side-point, but I think the authors do not understand the term “static dipoles” vs. dynamic dipoles. It does not mean fixed dipoles like e.g. force fields. In this context, the choice of the authors indeed corresponds to static dipoles, please refer to Pasquarello et al. 10.1103/PhysRevB.68.174302.”

Authors’ reply: We thank the Reviewer for the semantic clarification.

Reviewer #1 (Remarks to the Author):

This will be my third review of the present manuscript and I begin to believe that I understand the difficulties with the presentation. Adding an electric field to act on an otherwise equilibrated simulation is of course a perturbation that drives the system to a new equilibrium. Until this is reached the system is out of equilibrium and of course various properties will show a drift. Typically, unequilibrated simulations are not reported in the MD community and to discuss the results in terms of length of the simulations and only using the last part of the trajectory versus averaging over the whole trajectory then becomes rather misleading.

I recommend that the authors rather focus on the time-scale to reach equilibrium after the field has been turned on. Focus on the non-equilibrated character of shorter trajectories! It is clear to anyone in the MD field that, if one starts averaging a property from the initial guess without proper equilibration then the results will be very different from averaging over the final part of the trajectory. This seems to be part of the issue with the present paper - that the authors focus too much on the averaging rather than the equilibration.

I believe that the authors are aware of this equilibration aspect - it's a good motivation for running long trajectories, but they don't make this aspect clear (enough). The discussion about averaging over the entire trajectory versus the final part becomes rather confusing when one realizes the non-equilibrated character of the simulation.

I recommend the authors to clearly state that the system takes significant time to equilibrate and that this is the reason for the long trajectories and for only using the part of the trajectories that they consider to be equilibrated when analyzing the properties of the system. This is normal procedure. The whole discussion about using the whole or last part of the trajectory then becomes a moot point - of course one first has to ensure an equilibrated system before discussing the properties of the system.

The approach to equilibrium after an external stimulus is of course also of interest. Several such properties are shown and are well-described and presented, but the authors could make a better presentation if the discussion about the length of trajectories and the averaging instead were focused on the aspect of equilibrating the sample rather than the obfuscating details about averaging. In fact, emphasizing the length of simulations required to reach an equilibrated system could be more of a novelty that justifies publication in Nature Communications.

In discussing this, the authors could benefit from the (seeming) equilibration times with different (constant) field strengths in Figure 1a of reference [38] (Elgabarty et al.) showing the time-scale to reach equilibrium in the shown property as function of field-strength. Of course, different properties equilibrate on different time-scales, as also discussed in the present manuscript in connection with the response of individual molecules compared to the more "sluggish" response of the extended H-bonded network.

I do believe that the present manuscript with the presented data has merit enough to be published in Nature Communications, but I strongly recommend the authors to carefully go through their manuscript taking the above general comments into account.

I have some additional more specific comments below. If the authors make a serious consideration of the above points (if they agree they are valid), which I believe would make the message clearer, and take into account the specific comments below - then I would recommend it be published.

Specific comments:

- 1) Abstract: I recommend adding "bulk" to read "...first evidence of electrofreezing of bulk liquid water..." to make it a little less broad and general (and distinguish from surface-induced electrofreezing).
- 2) Page 2, line 7. I suggest to remove "new" to read "...transition to a ferroelectric glassy state..."
- 3) Page 7, line 5. Water "ionization". Ionization typically means removal of an electron. I assume the authors mean dissociation into OH^- and H_3O^+ ?
- 4) Page 7, lines 6-7. I don't believe "windowing" is the issue here. It just takes a long time for the system to reach this state.
- 5) Page 9, last two lines. "Such time-dependence, key in...". The authors have simply realized that it takes a long time to equilibrate the new state, and since equilibration is always central to any thermodynamical/structural sampling, it could actually be viewed as a natural thing to do - not really "key".
- 6) Page 10, middle. I suggest to add "near-" to read "near-complete freezing" since there seems to be some residual MSD.
- 7) Page 10, added text starting with: "We emphasize...". I would suggest to remove it. Of course, for an equilibrated system it makes no difference if the sampling is done far apart - it is just inefficient. The issue here is not the sampling, but the equilibration.
- 8) Page 12, second paragraph. I would recommend to change the focus from "isolating late portions.." to simply state that "The new f-GW phase takes a long time to reach and equilibrate and thus requires long simulations".
- 9) Page 12, third paragraph. Add "as" to read "- such as those established...".
- 10) References. Will likely be fixed but references 14 and 24 have some problems.

Reviewer #1

Query: “I recommend that the authors rather focus on the time-scale to reach equilibrium after the field has been turned on. Focus on the non-equilibrated character of shorter trajectories! It is clear to anyone in the MD field that, if one starts averaging a property from the initial guess without proper equilibration then the results will be very different from averaging over the final part of the trajectory. This seems to be part of the issue with the present paper - that the authors focus too much on the averaging rather than the equilibration.

I believe that the authors are aware of this equilibration aspect - it's a good motivation for running long trajectories, but they don't make this aspect clear (enough). The discussion about averaging over the entire trajectory versus the final part becomes rather confusing when one realizes the non-equilibrated character of the simulation.

I recommend the authors to clearly state that the system takes significant time to equilibrate and that this is the reason for the long trajectories and for only using the part of the trajectories that they consider to be equilibrated when analyzing the properties of the system. This is normal procedure. The whole discussion about using the whole or last part of the trajectory then becomes a moot point - of course one first has to ensure an equilibrated system before discussing the properties of the system.

The approach to equilibrium after an external stimulus is of course also of interest. Several such properties are shown and are well-described and presented, but the authors could make a better presentation if the discussion about the length of trajectories and the averaging instead were focused on the aspect of equilibrating the sample rather than the obfuscating details about averaging. In fact, emphasizing the length of simulations required to reach an equilibrated system could be more of a novelty that justifies publication in Nature Communications.

In discussing this, the authors could benefit from the (seeming) equilibration times with different (constant) field strengths in Figure 1a of reference [38] (Elgabarty et al.) showing the time-scale to reach equilibrium in the shown property as function of field-strength. Of course, different properties equilibrate on different time-scales, as also discussed in the present manuscript in connection with the response of individual molecules compared to the more “sluggish” response of the extended H-bonded network.

I do believe that the present manuscript with the presented data has merit enough to be published in Nature Communications, but I strongly recommend the authors to carefully go through their manuscript taking the above general comments into account.”

Authors' reply: We thank the Reviewer for this insightful report, that we have followed and adopted to carefully revise our manuscript. The revised article now explicitly emphasizes that the equilibration time is in the order of 150 ps, a timescale never reached before with *ab initio* simulations. The revised manuscript also emphasizes that the results there presented are obtained upon running analysis on equilibrated trajectories, which reach up to 500 ps. In the manuscript, we now emphasize that reported results are at equilibrium conditions. We have added the following sentences to the Introduction:

The application of intense EFs to liquid water induces a fast response of water molecules which align their dipole parallel to the field direction [38]. On the other hand, the intrinsic cooperativity of HBs acts as a competing force, slowing down the relaxation and retarding the equilibration times. In this work, we model the response of water to external EFs in the range [0.05 – 0.15] V/Å. We perform extensive AIMD simulations and show that

equilibration requires ~ 150 ps, while previous AIMD studies reached 100 ps or less [38, 39]. We show that, at ~ 150 ps, fields of $0.10 \leq E_f \leq 0.15$ V/Å induce a transition to a ferroelectric glassy state that we call f-GW (ferroelectric glassy water). The transition is signaled by the freezing of translational and rotational degrees of freedom, the suppression of the fluctuations of the HBN, and a drop in the potential energy. Fields of the order of 0.05 V/Å, on the other hand, are not strong enough to induce electrofreezing and the system remains in the liquid phase, although characterized by very sluggish dynamics.

Moreover, the following sentence now opens the Results Section:

As mentioned in the previous Section, equilibration upon the application of external EFs of $0.05 - 0.15$ V/Å is achieved after ~ 150 ps. In this Section, we present results obtained at equilibrium conditions, unless otherwise specified.

As the Reviewer and the Editorial team recognize, the study of the evolution of the HBN topology is novel and insightful. Therefore, we have kept a short, but fully explanatory, description of the choice of 50 ps time windows, while removing other unnecessary mentions to time windows.

The manuscript now clearly states that equilibration time is in the order of 150 ps, a timescale never reached before with ab initio simulations. As the Reviewer correctly reports, different quantities require different times to relax. To follow up on this, we now report, in the Supplementary Information, two new plots that prove that structural and topological quantities adopted in this work are fully relaxed (relaxation on dynamical quantities was already reported and discussed in the previous versions of the manuscript). We have computed differences, between consecutive time windows, of the radial distribution functions (figure S5) and of the number of n-member rings (figure S13). Figure S5 shows that structural relaxation occurs on time scales of ~ 100 ps for the lowest field (0.05 V/Å), and ~ 150 ps for the higher fields (0.10 and 0.15 V/Å). We have added the following comments on the Results Section:

To verify that the systems reach structural equilibrium at the times mentioned above, we compute the difference between $g_{OO}(r)$ s at consecutive time windows. As shown in Fig. S5-a, the field 0.05 V/Å induces structural equilibrium after ~ 100 ps. On the other hand, structural equilibration occurs after ~ 150 ps for higher fields, as shown in Fig. S5-b and Fig. S5-c.

Figure S13 shows that topological relaxation occurs on the same timescales. We have added the following comments of the Results Section:

Our investigations on the time evolution of the HBN exposed to external EFs indicate that the systems reach equilibrium configurations after ~ 150 ps. To confirm this, we compute the difference, at consecutive time windows, of the topology of the HBN. In Fig. S13 we report the difference of n-member rings, normalized for the number of water molecules, between consecutive time windows. The upper panel displays the results for 0.05 V/Å and shows that differences are attenuated at consecutive time windows, but never wear out since the system is still in the liquid phase. The middle and the lower panels, on the other hand, display the results for 0.10 V/Å and for 0.15 V/Å, respectively, and clearly show that the HBN converges to steady configurations after ~ 150 ps.

Query: “I have some additional more specific comments below. If the authors make a serious consideration of the above points (if they agree they are valid), which I believe would make the message clearer, and take into account the specific comments below - then I would recommend it be published.

Authors' reply: *Reviewer's suggestions were extremely valid, and we followed them diligently. Shifting the attention from the sampling to the equilibration time (as well as all other minor suggestions) indeed strongly improved the manuscript.*

Reviewer #1 (Remarks to the Author):

The authors have overall addressed my comments and followed my suggestions. However, there seems to be a confusion between equilibration and being in the liquid phase that could be a bit of an embarrassment to the authors if not removed (points 1 and 2 below). This is easily addressed and I point out where and (more or less) how below.

More importantly, the issue whether the system equilibrates more rapidly or more slowly at lower field-strength needs to be made more clear. Elgabarty et al. seem to indicate more rapid equilibration at higher field-strength. The present work seems to propose the opposite or not be clear about this aspect.

If the authors address the now minor points below and add a brief discussion/comment on relaxation time dependence on field-strength (comparing also with Elgabarty fig. 1a), then I recommend publication (finally).

1) Page 9, added text, bottom. The comment on the behavior of the system exposed to 0.05 V/\AA needs some reflection on the part of the authors. Specifically, “and shows that differences are attenuated at consecutive time windows, but never wear out since the system is still in the liquid phase.” This confused the present reviewer. However, I finally realized that the authors, instead of “differences are attenuated” (probably) meant to say that “fluctuations are attenuated”. Differences imply poor equilibration while fluctuations do imply a soft matter system (normally, a liquid is characterized by diffusion).

2) Page 12, end first paragraph. The statement that: “Overall, the inspected quantities computed within the last ~ 150 ps of simulation are stable in time, indicating that the system is genuinely a liquid” should furthermore be reconsidered since properties being stable in time has nothing to do with the aggregation state (gas, liquid, solid) but how well the system is equilibrated. A liquid is differentiated from a solid, e.g., by the diffusivity - not by properties being stable in time.

3) SI, page 10. Again, there is a reference to the “liquid state” in “Such difference does not disappear for 0.05 V/\AA because the system is in the liquid phase”. Please consider changing difference to *fluctuations*!!

4) Page 3, final paragraph of the introduction. The statement that “...an issue particularly relevant when the system is drifting out of equilibrium.” is inconsistent with the aspect of equilibration. The system is drifting *towards* a new equilibrium. This is a minor point (it is true that the system moves from the state that was equilibrated without the field, but that is not the target of the study - as I understand it), but could be helpful to put the mindset of the reader in the direction of equilibration.

Response to the Reviewer

The authors have overall addressed my comments and followed my suggestions. However, there seems to be a confusion between equilibration and being in the liquid phase that could be a bit of an embarrassment to the authors if not removed (points 1 and 2 below). This is easily addressed and I point out where and (more or less) how below.

We thank the Reviewer for their insightful report.

More importantly, the issue whether the system equilibrates more rapidly or more slowly at lower field-strength needs to be made more clear. Elgabarty et al. seem to indicate more rapid equilibration at higher field-strength. The present work seems to propose the opposite or not be clear about this aspect.

If the authors address the now minor points below and add a brief discussion/comment on relaxation time dependence on field-strength (comparing also with Elgabarty fig. 1a), then I recommend publication (finally).

We observe the same behaviour as Elgabarty et al. at the very short time scales inspected in that work. Our figure Supplementary 3 already shows this comparison. To make this clear, we have emphasized this point in the supplementary information as

Our results are in qualitative agreement with Ref. [5].

We mention “qualitative” because fig. 1a of Elgabarty reports results for three fields and only one is comparable with ours. In particular, the highest field reported in Elgabarty is 0.43 V/\AA which induces extensive proton dissociation within 10ps. Therefore, we are not sure what is the meaning of fig. 1a for this very high field.

We have also added the following sentence in the Results Section:

Our results show that equilibration is achieved after $\sim 150 \text{ ps}$ in the case of the strongest field of 0.15 V/\AA^{-1} , while it requires $\sim 200 \text{ ps}$ for the lower fields of $0.10 - 0.05 \text{ V/\AA}^{-1}$

1) Page 9, added text, bottom. The comment on the behavior of the system exposed to 0.05 V/\AA needs some reflection on the part of the authors. Specifically, “and shows that differences are attenuated at consecutive time windows, but never wear out since the system is still in the liquid phase.” This confused the present reviewer. However, I finally realized that the authors, instead of “differences are attenuated” (probably) meant to say that “fluctuations are attenuated”. Differences imply poor equilibration while fluctuations do imply a soft matter system (normally, a liquid is characterized by diffusion).

We have modified the manuscript as suggested by the Reviewer.

2) Page 12, end first paragraph. The statement that: “Overall, the inspected quantities computed within the last $\sim 150 \text{ ps}$ of simulation are stable in time, indicating that the system is genuinely a liquid” should furthermore be reconsidered since properties being

stable in time has nothing to do with the aggregation state (gas, liquid, solid) but how well the system is equilibrated. A liquid is differentiated from a solid, e.g., by the diffusivity - not by properties being stable in time.

We have modified the manuscript as suggested by the Reviewer. We have added the following sentence in the Discussion Section:

Overall, the inspected quantities computed within the last ~ 150 ps of simulation including active rotational and translational diffusivity indicate that the system is genuinely a liquid.

3) SI, page 10. Again, there is a reference to the "liquid state" in "Such difference does not disappear for 0.05 V/°A because the system is in the liquid phase". Please consider changing difference to fluctuations!!

We have modified the manuscript as suggested by the Reviewer.

4) Page 3, final paragraph of the introduction. The statement that "...an issue particularly relevant when the system is drifting out of equilibrium." is inconsistent with the aspect of equilibration. The system is drifting towards a new equilibrium. This is a minor point (it is true that the system moves from the state that was equilibrated without the field, but that is not the target of the study - as I understand it), but could be helpful to put the mindset of the reader in the direction of equilibration.

We have modified the manuscript as suggested by the Reviewer.